



# The role of household adaptation measures to reduce vulnerability to flooding: a coupled agent-based and flood modelling approach

Yared Abayneh Abebe[1, 2], Amineh Ghorbani[3], Igor Nikolic[3], Natasa Manojlovic[4], Angelika Gruhn[4] and Zoran Vojinovic[1, 5, 6, 7]

[1]Environmental Engineering and Water Technology Department, IHE Delft Institute for Water Education, Westvest 7, 2601 DA, Delft, The Netherlands
[2]Department of Biotechnology, Delft University of Technology, Van der Maasweg 9, 2629 HZ, Delft, The Netherlands
[3]Faculty of Technology, Policy and Management, Delft University of Technology, Jaffalaan 5, 2628 BX, Delft, The Netherlands.
[4]River and Coastal Engineering, Hamburg University of Technology, Denickestraße 22 (I), 21073 Hamburg, Germany.
[5]Center for Water Systems, College of Engineering, Mathematics and Physical Sciences, University of Exeter, EX4 4QF Exeter, UK.
[6]Faculty of Civil Engineering, University of Belgrade, Bulevar kralja Aleksandra 73, 11000 Belgrade, Serbia.
[7]Department of Hydraulic and Ocean Engineering, National Cheng Kung University, No. 1 University Road, Tainan, Taiwan.

*Correspondence to*: Yared Abayneh Abebe (y.a.abebe@tudelft.nl)

**Abstract.** Flood adaptation measures implemented at household-level play an important role in reducing communities' vulnerability. The aim of this study is to enhance the current modelling practices of human-flood interaction to draw new insights for FRM policy design. The paper presents a coupled agent-based and flood model for the case of Hamburg, Germany to explore how individual adaptation behaviour is influenced by flood events scenarios, economic incentives, and shared and

individual strategies. Simulation results show that a unique trajectory of adaptation measures and flood damages emerge from different flood event series. Another finding is that providing subsidies increases the number of coping households in the long run. Households' social network also has a strong influence on their coping behaviour. The paper also highlights the role of simple measures such as adapted furnishing, which do not incur any monetary cost, in reducing households vulnerability and preventing millions of euros of contents damage. Generally, we demonstrate that coupled agent-based and flood models can

potentially be used as decision support tools to examine the role of household adaptation measures in flood risk management. Although the findings of the paper are case-specific, the improved modelling approach shows the potential to be applied in testing policy levers and strategies considering heterogeneous individual behaviours.

## 1 Introduction

One of the goals of flood risk management (FRM) is evaluation of strategies, policies, and measures to foster flood risk

reduction and promote continuous improvement in flood preparedness and recovery practices (IPCC, 2014). As flood risk is a function of flood hazard and communities' exposure and vulnerability, one way of reducing flood risk is by reducing the vulnerability at the household level. Focusing on the physical and economic aspects, measures to reduce vulnerability include



elevating houses, retrofitting, dry/wet floodproofing, insurance and subsidies. These measures either prevent flooding or minimise the impact. While measures such as subsidies are offered by authorities or aid groups, the decision to implement

most adaptation measures is made at the household level.

Household adaptation behaviour is affected by many factors such as flood risk perception, experience with flooding, socioeconomic and geographic factors, reliance on public protection, and competency to carry out adaptation measures (Bubeck et al., 2012). The current literature mainly makes use of empirical research to draw insights on the role of household adaptation behaviour to reduce flood risk (for example, Botzen et al., 2019; Grahn and Jaldell, 2019; Grothmann and Reusswig,

2006; Poussin et al., 2014; Schlef et al., 2018). Nevertheless, modelling efforts that bring behavioural and physical attributes together can further enrich these insights and add even more knowledge by incorporating the complex reality surrounding the human-flood interactions.

One of the research gaps in the current literature that present models to study household flood adaptation behaviour (for example, Erdlenbruch and Bonté, 2018; Haer et al., 2016) is that flood events are not included in the simulation models. These

studies define flood experience as an agent attribute that is set initially and stays the same throughout the simulations. A household that was not flooded in past events may get flooded in the future and may re-evaluate previous adaptation decisions, which in turn necessitates that flood events are included in the modelling. The second gap is that the effects of an economic incentive on the adaptation behaviour of individuals have not been addressed in the models. Such analysis would provide an understanding of how much incentives contribute to flood risk reduction.

This study aims to enhance the current modelling practices of human-flood interaction to address the shortcomings of the current literature and draw new insights for FRM policy design. To achieve this aim, we build a coupled agent-based and flood models that comprehensively includes the human and the flood attributes in a holistic manner (Vojinovic, 2015). Agent-based models (ABMs) are computational models where autonomous and heterogeneous agents (for example, households) interact with each other and their environment (Railsback and Grimm, 2012), exploring the behaviour of agents in a system. The

coupled ABM-flood model builds on empirical and modelling insights in the literature: (i) by presenting an integrated simulation model instead of only agent-based models, and (ii) by testing the effects of economic incentives and institutional configurations that have not yet been studied in the context of household flood adaptation behaviour. We use the protection motivation theory (PMT) (Rogers, 1983) to investigate household-level decision making to adopt mitigation measures against flood threats.

More specifically, this paper extends two studies presented in (Birkholz, 2014) and (Abebe et al., 2019b). Birkholz qualitatively explored PMT to study household flood preparedness behaviour in the German city of Hamburg. Birkholz collected information on local communities' flood risk perceptions and flood preparedness using semi-structured interviews. The current study uses the qualitative study as a base to conceptualise and further explore the household flood preparedness behaviour in Hamburg using an ABM. Abebe et al. (2019b) employ the coupled flood-agent-institution modelling (CLAIM) framework

developed in (Abebe et al., 2019a) to conceptualise the agent-flood interaction by decomposing the system into five components – agents, institutions, urban environment, physical processes and external factors. Their main focus was to study





the implications of formal rules as institutions. In contrast, the current study mainly investigates the effect of informal institutions in the form of shared strategies applying the CLAIM framework. Additionally, the study examines individual strategies that affect households' adaptation behaviour.

The remainder of the paper is structured as follows: Section 2 describes the study area. Section 3 provides a brief description of PMT and explains how it is conceptualized for the study area. Section 4 discussed how CLAIM is used to decompose the system, the ABM and flood model setups, model evaluations and experimental setups. Section 5 presents the results of the modelling exercises, followed by a discussion of the implications of the study findings and conclusions in Section 6.

## 2 Study area

We develop a coupled ABM-flood model that uses PMT as a tool to model households' flood vulnerability reduction behaviour for the FRM case of Wilhelmsburg, a quarter of Hamburg, Germany. The Wilhelmsburg quarter is built on a river island formed by the branching River Elbe, as shown in Figure 1. Most areas in Wilhelmsburg are just above sea level. Thus, flood defence ring of dykes and floodwalls protect the quarter. In 1962, a hurricane-induced storm surge (5.70 m above sea level) overtopped and breached the dykes, and more than 200 people lost their lives and properties were damaged due to coastal

flooding in Wilhelmsburg (Munich RE, 2012). As a result, the authorities heightened and reinforced the coastal defence system. According to the Munich RE report, after 1962, eight storm surges of levels higher than 5.70 m occurred (most between 1990 and 1999), but none of the events caused any damage as coastal protection has been improved.

Those events reminded residents of the potential risks of coastal flooding, while, at the same time, increasing their reliance on the dyke protection system. The reliance on public protection is promoted by the authorities, who do not encourage the

implementation of individual flood risk reduction measures referring to the strength of the dyke system. On the other hand, the authorities disseminate warning and evacuation strategies to the public, acknowledging that there can be a flood in future. There is a probability that a storm surge bigger than the design period of the coastal defence may occur in the future, and climate change and sea level rise may even intensify the event. Hence, protecting houses from flooding should not necessarily be the responsibility of the authorities. Households should also have a protection motivation that leads to implementing

measures to reduce flood risk.

## 3 Protection motivation theory

As shown in Figure 2, PMT has three parts – sources of information, cognitive mediating processes and coping modes (Rogers, 1983). The *sources of information* can be environmental such as seeing what happens to others and intrapersonal such as experience to a similar threat. Triggered by the information, the *cognitive mediation process* includes the threat and coping

appraisals. The *threat appraisal* evaluates the severity of and the vulnerability to the threat against the intrinsic and extrinsic positive reinforcers. The *coping appraisal* evaluates the effectiveness of an adaptation measure to mitigate or reduce the risk,



the ability to implement the measure, and the associated cost to implement the measure. If the threat and coping appraisals are high, households develop a *protection motivation* that leads to action. The *coping modes* can be a single act, repeated acts, multiple acts or repeated, multiple acts.

Originally developed in the health domain (Rogers, 1983), PMT has been extended and applied in diverse domains that involve a threat for which individuals can carry out an effective recommended response available (Floyd et al., 2000). For example, in FRM studies, Poussin et al. (2014) extended the PMT by adding five factors – flood experience, risk attitudes, FRM policies, social networks and social norms, and socioeconomic factors – that directly determine the protection motivation of households. Two studies applied PMT in ABMs to test the effectiveness of flood risk communication strategies and the influence of social

network on the adoption of protective measures to reduce households' vulnerability to flooding (Erdlenbruch and Bonté, 2018; Haer et al., 2016). They compute the odds ratio and probability of implementation to model household decision on flood preparedness. One of the conclusions of the studies is that communication policies should have information regarding both the flood threat and coping methods to increase the adaptation rate.

**Conceptualizing the protection motivation theory for Wilhelmsburg**

In the present work, we have modified the original PMT (Rogers, 1983) to use it in an FRM and ABM contexts for the specific case of Wilhelmsburg. In the original theory, the sources of information initiate both the threat appraisal and coping appraisal processes. However, in the current study, the sources of information influence the threat appraisal only. We assume that if there is a threat and need to implement a coping measure, the agents know the type of measure they implement based on their

house categories (see Table 1).

In the threat appraisal, the *maladaptive response* is the current behaviour of not implementing household-level flood vulnerability reduction measures. In the case of Wilhelmsburg, the maladaptive response is affected by flood experience, reliance on public protection (i.e., the dyke system), climate change perception and source of information.

- The flood experience refers to any experience from which households can be directly affected by flooding, or they
have witnessed flooding that affected others in Wilhelmsburg.

- The reliance on public protection is related to the flood experience. Residents of Wilhelmsburg who have not experienced flooding have a high reliance on the dyke system. The fact that seeing the dykes on a daily basis give residents a sense of protection and underestimate the flood threat. The reliance on public protection is also associated with the trust the residents have on the authorities when it comes to FRM. However, as some informants who
experienced the 1962 flood described, the reliance on the dyke system drops if flooding occurs in the future (Birkholz, 2014).

- We include agents' climate change perception as a factor as some residents of Wilhelmsburg described that sea level rise might increase the occurrence of flooding in future. The effects of climate change create some discomfort and stress, and hence, it is seen as a source of concern. Besides, Germans, in general, are concerned about climate change
in which 86% are "extremely to somewhat worried" (NatCen Social Research, 2017).

- The source of information is an important factor that shapes residents perception of flood risk. The municipal and state authorities have a firm belief that the dyke system is the primary flood protection measure, and there is no need





to implement individual measures to protect properties. However, these authorities communicate evacuation strategies in case the dykes fail or overtopped by a storm surge. On the other hand, other sources such as experts from the Technical University of Hamburg-Harburg organised flood risk awareness workshops presenting the flood risk in Wilhelmsburg and different adaptation measures that individuals could implement. Media also has a role in creating concern by showing flooding and its impacts in other German cities and even other countries.

In the coping appraisal, the *adaptive response* is developing a protection motivation behaviour to implement flood vulnerability reduction measure. The factors that affect the response probability in this conceptualization are personal flood experience, house ownership, household income, subsidy from the state and social network.

- Personal flood experience refers to a direct flood experience in which an agent's house was flooded before. It is a major factor that drives the adaptive response (Bubeck et al., 2012). The factor is used as a proxy for behaviours in case of near-miss flood events as agents tend to make riskier decisions if they escape damage while others are flooded (Tonn and Guikema, 2017).

- We include house ownership as a factor though it has a small to medium effect on the adaptive response (Bubeck et al., 2012). However, this factor is also used as a proxy for tenancy, which is an important factor since tenants tend not to implement measures. Hence, house ownership in this context specifies whether an owner or a tenant occupies a house at a given time.

- Household income has a significant influence on the adaptive response especially when agents implement measures that bring structural changes or adjustments to buildings such as flood proofing and installing utility systems to higher ground (Bubeck et al., 2013). Hence, this factor affects only those households that intend to implement "structural measures".

- The subsidy is any financial help the authorities may provide to encourage implementation of individual adaptation measures. Currently, the authorities do not provide subsidies as they invest only on public protections. But, the assumption is that if a future low probability storm surge overtop or overflow the dyke system and flooding occurs, the authorities may take responsibility for the damages of properties given their assurance that people are safe and do not need to implement individual measures. As the subsidy is financial support, we conceptualise this factor similar to the household income affecting household agents that implement structural measures.

- The social network factor represents agents' relatives, friends or neighbours who have implemented any adaptation measure. Bubeck et al. (2013) showed that residents conform to the protection mitigation behaviour of others in their social network.

The state subsidy and the household income are proxy measures for the financial *response cost* of implementing the measures. In terms of other costs such as time and effort, we assume that the agents have no limitation. The assumptions related to response efficacy is that agents implement the adaptation measure specified in the shared strategy based on the type of houses they own, and the measure is assumed to be effective to reduce flood damage. However, it does not necessarily imply that the measure is the best possible. Similarly, the assumption related to self-efficacy is that either agents need to hire technicians that are capable of successfully implementing the measures or they are capable of implementing the measures by themselves. Appendix A lists the assumptions made to conceptualise and develop the model.





At last, *protection motivation* is an intention to implement coping responses (Rogers, 1983), which may not necessarily lead to actual behaviour (Grothmann and Reusswig, 2006). In our conceptualisation, agents may delay the implementation of measures after they positively appraise coping. Agents may also change their behaviour through time and abandon temporary measures affecting their protection motivation.

## 4 CLAIM decomposition and model setups

We use the CLAIM framework (Abebe et al., 2019a) to decompose and structure the FRM case of Wilhelmsburg as CLAIM provides the means to explicitly conceptualise household behaviour and decision making, households interaction among themselves and with floods, and institutions that shape household behaviour. CLAIM has five components: (i) *Agents* are entities that represent an individual or composite actors/stakeholders in a model; (ii) *Institutions* are the rules, norms and strategies defined by actors to organize their actions, interactions and decision makings (Crawford and Ostrom, 1995); (iii) the *urban environment* is where agents live and floods occur, and is the component that connects the human and flood subsystems; (iv) *physical processes* are hydrologic and hydrodynamic components related to floods; (v) *external factors* are elements that affect the "local" agent-flood interactions but are not affected by the direct actions and interactions of agents in the local settings. Agents and institutions are part of the human subsystem and are modelled using ABMs, whereas the physical processes are part of the flood subsystem and are modelled using hydrodynamic models. As the urban environment links the two subsystems, features of this component can be conceptualised in either of the subsystems. Similarly, external factors may affect both subsystems, and hence, can be part of either subsystems. The conceptualisations of the CLAIM components are discussed in the following two subsections.

The primary source of data for the conceptualisation is the doctoral dissertation by Birkholz (2014). Birkholz applied semi-structured, in-depth interviews with residents; academic and grey literature reviews; and personal observation of the study area. Besides, we use local expert knowledge of the study area to develop the conceptual model.

### 4.1 Agent-based model setup

We will describe the FRM case of Wilhelmsburg using agent, institution, urban environment components of the CLAIM framework.

**Agents:** we identified two types of agents – the household and the authority agents.

- The household agents are representations of the residents of Wilhelmsburg. These agents live in residential houses. The actions they pursue include appraising threat and coping, implementing adaptation measure, and assessing direct damage. The agent attributes related to threat appraisal are flood experience, reliance on public protection, perception of climate change and source of information about flooding. The attributes related to coping appraisal are direct flood experience, house ownership and household income. If agents decide to implement an adaptation measure, they know which measure to implement based on the institutions identified. The conceptualization excludes businesses, industries, farmlands and other auxiliary buildings due to lack of data.





- The authority agent represents the relevant municipal and state authorities that have the mandate to manage flood risk in Wilhelmsburg. This agent does not have a spatial representation in the ABM. The only action of this agent is to provide subsidies to household agents based on the policy lever defined in the experimental setup of the ABM. We model subsidy in a more abstract sense that if agents receive a subsidy, they implement an adaptation measure assuming that agents are satisfied with the amount they receive.

**Urban environment**: The Wilhelmsburg quarter that is surrounded by the ring of dykes and walls defines the urban environment (see Figure 1). The household and authority agents live and interact in this environment. In our conceptualisation, we focus only on household behaviour to protect their houses. Therefore, the only physical artefacts explicitly included in the conceptual model are residential houses, which spatially represent the household agents in the ABM. They have geographical location represented using polygon features, as illustrated in Figure 1. These polygons are used to compute the area of the houses. Houses also have types, which are classified based on "the type of building, occupancy of the ground floor and the type of facing of the building." (Ujeyl and Rose, 2015, p.1540006–6). This study includes 31 types of houses, which we group into five categories: single-family houses, bungalows, IBA buildings, garden houses, and apartment/high-rise buildings. Appendix B provides a complete list of the 31 types of houses.

If a house is flooded, the potential building and contents damages of the house are computed in monetary terms based on the house type. A raster file represents the urban environment, and if floods occur, agents obtain information about flood depth at their house from the environment. The adaptation measures that households may implement do not have physical representations in the model though their impact is implicitly evaluated if a house is exposed to flooding.

**Institutions**: In Wilhelmsburg, there is a common understanding that it is the responsibility of the authorities to protect the people. There is no institution, formal or informal, that influence household behaviour to reduce vulnerability. As a result, we will test hypothetical shared strategies that may have some effect on household agents flood risk. The conceptual model consists of five institutions in which one is related to the authority agent providing subsidies to household agents and the rest related to households implementing vulnerability reduction measures depending on the house categories.

Institutions in CLAIM are coded using the ADICO grammar, which refers to the five elements institutional statements might contain: Attributes, Deontic, aIm, Condition and "Or else" (Crawford and Ostrom, 1995). Table 1 shows the five institutional statements that influence the implementation of individual flood risk reduction measures. When an agent is permitted to do an action (deontic *may*) with no explicit sanction (no "or else") for failing to do the action, the statement is referred to as a *norm*. In this case, the last institutional statement related to the subsidies is conceptualised as a norm. The authority agent may give subsidies, but it is not obliged to do so and faces no sanction if it decides not to provide subsidies. When the deontic and "or else" components are absent from an ADICO statement, the statement is referred to as *shared strategy*. Therefore, the first four statements in Table 1 are shared strategies as there are no sanctions for non-compliance with the statements (no "or else" component), and there are no deontic. When a shared strategy drives a system, agents do what the majority in that system does. As a result, a household implements a measure when the majority of households implement the adaptation measure. However, the household also has the option not to implement the measure without incurring any punishment.





In our conceptualisation, households implement a specific primary measure or a secondary measure (stated in the "aim") based on the category of a house they occupy (stated in the "condition"). Considering primary measures, as most single-family houses in Wilhelmsburg have two or three floors, household agents that live in such houses install utilities such as heating, energy, gas and water supply installations in higher floors. Household agents that live in bungalows and IBA buildings implement flood adapted interior fittings such as walls and floors made of waterproofed building materials. Agents that live in garden houses and apartment/high-rise buildings implement flood barriers. The barriers implemented by garden houses are sandbags and water-tight windows and door sealing while the latter implement flood protection walls. Household agents that have already implemented a primary measure may also implement a secondary measure. This measure is adapted furnishing, which includes moving furniture and electrical appliances to higher floors. As most bungalows and garden houses are single-storey housings, they do not implement adapted furnishing.

Installing utilities in higher floors and flood adapted interior fittings are permanent measures that alter the structure of the house, and we assume that once they are implemented, they will not be abandoned. Therefore, in PMT terminology, implementing these measures is a *single act coping mode*. In contrast, flood barriers and adapted furnishing are temporary measures in which agents must decide whether to implement them every time, just before a flood event. Therefore, implementing these measures is a *repeated acts coping mode*. Implementing both primary and secondary measures is a *repeated, multiple acts coping mode.*

**External factors**: There is no external institution conceptualised in this model. Although there is a European Union Floods Directive that requires member states such as Germany to take measures to reduce flood risk, it does not specify the type of measure implemented. In Wilhelmsburg, the authorities invest primarily on the dyke system; hence the implications of the Floods Directive on individual adaptation measure is not relevant in this study.

**Dynamics of the agent-based model**

The model implementation flow chart shown in Figure 3 lays out the actions agents perform at every time step. First, household agents assess if they perceive flood as a threat. If they do, they appraise coping that leads to protection motivation behaviour. Second, if there is the intention to implement a measure, they implement the adaptation measure specified in the institutional table. Lastly, if there is a flood event at a given time step, the house layer is overlaid with a flood map corresponding to the event. Households check the flood depth at their property and assess the building and contents damages. Agents' attributes are updated if the actions change their states. This process is performed until the end of the simulation time. We will describe below how the actions – threat appraisal, coping appraisal, adaptation measure implementation, damages assessment and measures abandoning – are evaluated in the model.

**Action 1: threat appraisal**





In the ABM, the factors that affect household agents perception of flood threat in Wilhelmsburg are their flood experience
($FE$), their reliance on public protection ($R$), mainly the ring of dykes, their perception of future climate change ($CC$) and their
source of information ($SoI$). Household agents update the four factors every time step based on the following criteria:

o    $FE$ is related to whether an agent lives in Wilhelmsburg when a flood event happens, and it has a binary value of *Yes* and
*No*. The value of $FE$ changes only after a flood event as given in **Eq 1**. We assume that the flood experience does not
fade over time.

$$FE = \begin{cases} Yes, & if\ agent\ lives\ in\ Wilhelmsburg\ when\ flood\ occurs \\ No, & otherwise \end{cases} \qquad \text{Eq 1}$$

o    $R$ has a value of *Low*, *Medium* and *High*. It is dependent on $FE$ and whether an agent has direct flood experience (see **Eq
2**). The *Medium* value reflects the uncertain position of agents towards the dyke system if they witness flooding in
Wilhelmsburg. The value of $R$ does not change unless there is a flood event and agents are flooded. This attribute is
initialized based on the agents FE status.

$$R = \begin{cases} Low, & if\ FE = Yes\ \&\ agent\ is\ flooded \\ Medium, & if\ FE = Yes\ \&\ agent\ is\ not\ flooded \\ High, & if\ FE = No \end{cases} \qquad \text{Eq 2}$$

o    $CC$ has a value of *Yes*, *No* and *Uncertain*. The $CC$ value of every agent is generated randomly from a uniform distribution,
as shown in **Eq 3**. The value of this attribute may change over the simulation period. Assuming that agents may update
their $CC$ attribute at least once every $Y_{CC}$ years, there is a probability of $1/Y_{CC}$ at every time step to update the attribute
using **Eq 3**.

$$CC = \begin{cases} Yes, & if\ Random{\sim}U(0,1) \leq 0.44 \\ Uncertain, & if\ 0.44 < Random{\sim}U(0,1) \leq 0.86 \\ No, & if\ Random{\sim}U(0,1) > 0.86 \end{cases} \qquad \text{Eq 3}$$

o    We broadly categorize $SoI$ as "*information from Authorities,*" which informs agents that the dykes will protect everyone
and there is no flood threat, and "*information from other sources,*" which informs agents that there can be a flood threat
and agents need to prepare. $SoI$ is assigned to agents randomly. Similar to the $CC$ attribute, there is a probability of $1/Y_{SoI}$
to update the $SoI$ attribute assuming that agents may update this attribute at least once every $Y_{SoI}$ years.

The flood threat is a function of the four factors and agents assess their perception of flooding as a threat using a rule-based
decision tree (see Figure 4). If an agent has no experience of flooding, its reliance on public protection is high, and it perceives
no threat of flooding regardless of the other factors. On the other hand, if an agent has low reliance on the dyke system, it
perceives flooding as a threat regardless of the other factors. In case an agent's reliance on public protection is intermediate,
its perception of climate change determines the threat appraisal. A concern regarding future impacts of climate change results
in a perception of flood threat while no concern leads to no perception of the flood threat. If an agent is uncertain about climate
change impacts, its source of information determines the threat appraisal. As some of the attributes of agents may change over
time, all agents appraise threat at every time step.

**Action 2: coping appraisal**



Coping behaviour is initiated depending on agents' belief in their ability to implement a measure, agents' expectation that the measure removes the threat or improves the situation, and the perceived costs of implementation. In our model, coping appraisal is influenced by agents direct flood experience, i.e., if they had personal flood experience ($PFE$), house ownership ($HO$), household income ($HI$), state/government subsidy ($SS$) and the number of measures within agent's social network ($SN$).

o   $PFE$ has a value of *Yes* or *No* based on agents direct flood experience. This attribute is initialised as *No* for all agents. The value of $PFE$ changes only when an agent's house is flooded after an event as given in **Eq 4**.

$$PFE = \begin{cases} Yes, & if\ agent\ has\ direct\ flood\ experience \\ No, & otherwise \end{cases}$$   Eq 4

o   $HO$ has a value of *Own* or *Rented*. Initially, agents are randomly assigned to one of the values. Then, we assume that the house ownership of a percentage of the household agents changes randomly, at every time step.

o   $HI$ has a value of *Low* or *High*. Similar to the house ownership, we assume that the income of a percentage of the household agents changes randomly, at every time step. It should be noted that this factor affects the agents that implement permanent adaptation measures of installing utilities in higher storeys and flood adapted interior fittings, which are classified as structural measures (see Bubeck et al., 2013, p.1330).

o   $SS$ has a value of *Yes or No*. This variable is related to the last institution mentioned in Table 1. In the ABM setup, it is used as a policy lever to test the effect of subsidies on the implementation of structural adaptation measures.

o   $SN$ has a value of *Low* or *High*. As shown in **Eq 5**, this factor depends on the number of agents that implement a specific type of adaptation measure for a given house category. If the number is greater than a threshold, agents who occupy that same house category will have *High SN* value. Otherwise, $SN$ is *Low*.

$$SN = \begin{cases} High, & if\ NA_{measureType} \geq threshold \\ Low, & otherwise \end{cases}$$   Eq 5

where, $NA_{measureType}$ is the number of agents that implement a specific measure type depending on the category of house they occupy.

Coping is a function of the five factors, and agents appraise their coping using a rule-based decision tree illustrated in Figure 5. For households that implement a structural measure, the full decision tree is evaluated while for those that implement non-structural measures (i.e., temporary measures), shapes and lines in dashed line are not assessed. If household agents have direct flood experience, the conditions that they would not intend to cope and implement a structural measure are if they occupy a rented house and (i) they have high income but have low $SN$, or (ii) they have low income and received no subsidy, or (iii) they have low income and received a subsidy, but have low $SN$. If agents live in their own house, the only condition that they would not intend to cope is if they have low income, received no subsidy and have low $SN$. In all the other cases, agents coping appraisal results in intention to cope. If agents do not have direct flood experience, the only case that they develop a coping behaviour is when the agents own the house they occupy and (i) they have high income and have high $SN$, or (ii) they have low income, have received a subsidy and have high $SN$. In the rest of the cases, household agents do not develop coping behaviour.





In the case of household agents that implement temporary measures, if the agents have direct flood experience, the only condition that they would not intend to cope is if they occupy a rented house and have low $SN$. If agents do not have direct flood experience, the conditions that they would not intend to cope are: (i) if they occupy a rented house and (ii) if they own
the house but have low $SN$. In the rest of the cases, household agents develop coping behaviour.

An important aspect regarding the $SN$ factor is that as its value changes for all agents in the same house category and since all the agents may not follow the behaviour of other agents in their social network, we introduce a *shared strategy parameter* that works in tandem with the $SN$. If agents $SN$ factor is *High*, they develop a coping behaviour when a randomly drawn number from a uniform distribution is less than or equal to the *shared strategy parameter*.


**Action 3: household adaptation measure implementation**

Following Erdlenbruch and Bonté (2018), we introduce a *delay parameter* that affects measures implementation. The delay parameter represents the average number of years agents take to transform a protection motivation behaviour into an action, which is implementing a primary measure. The probability that a motivated individual will adapt at a given year is computed
as $p_{adaptation,primary} = 1/delay\ parameter$. We also introduce a *secondary measure parameter* that determines whether agents implement secondary measures. This parameter is set as a threshold value defined by the modeller's estimation. As shown in Figure 3 (b), agents consider implementing secondary measures only if they implement primary measures. The assumption is that those agents have already appraised coping positively, and they may have a protection motivation to implement a secondary measure. As stated earlier, only multi-storey house categories implement secondary measures.


**Action 4: damages assessment**

The impacts of a flood event can be estimated by the direct and indirect damages of flooding on tangible and intangible assets. In this study, we measure the flood impact based on the potential direct damages which are caused by the physical contact of floodwater with residential houses. We estimate the building and contents damages using depth-damage curves developed for
the 31 types of houses in Wilhelmsburg, as discussed in (Ujeyl and Rose, 2015). The building damages are related to replacement and clean-up costs, whereas the contents damages are related to replacement costs of fixed and dismountable furnishing. Figure 6 shows the depth-damage curves for the different house types.

If household agents implement adaptation measures, the building and contents damages of their house reduce. Based on empirical researches (Kreibich and Thieken, 2009; Poussin et al., 2015), we compute the damages reduced as a percentage
reduction of the ones presented in Figure 6. Installing utilities in higher storeys reduces the building damage by 36 while it has no impact on the contents damage reduction. Implementing flood adapted interior fittings reduces both damages by 53%. Implementing adapted furnishing reduces the contents damage by 77% while it has no impact on the building damage reduction. In the case of flood barriers, implementing sandbags and water-tight windows and door sealing reduces only the building damage by 29% whereas implementing flood protection walls reduces the flood depth by a maximum of one meter.






**Action 5: measures abandoning**

We also introduce an *adaption duration parameter* factor that affects measures abandoning, following Erdlenbruch and Bonté (2018). The adaptation duration parameter represents the average number of consecutive years a household agent implements an adaptation measure. It is used to estimate the probability that an agent abandons the measure at a given year. The likelihood

that a motivated individual abandons a measure at a given year is computed as $p_{abandoning} = 1/adaptation\ duration\ parameter$. This parameter affects only agents that implement temporary measures. The minimum adaptation duration would be one year. As shown in Figure 3 (c), we limit the frequency of abandoning a measure by an agent using the *abandoning frequency threshold*. The assumption is that agents will not abandon a measure any more if they abandon and implement it a certain number of times specified in the threshold. If an agent has implemented a secondary measure, the

first option to abandon is that measure. Otherwise, the agent abandons the temporary primary measure. In the latter case, the agent appraises coping once again.

Once the conceptual model is developed, we convert it to a programmed model using the Java-based Repast Simphony modelling environment (North et al., 2013). The ABM software developed in this study, together with the ODD protocol (Grimm et al., 2010) that describe the model, is available at https://github.com/yaredo77/Coupled_ABM-

Flood_Model_Hamburg.

**4.2 Flood model setup**

**Hydrologic and hydrodynamic processes**: Located in the Elbe estuary, the main physical hazard that poses a risk on Wilhelmsburg is storm surge from the North Sea. If the surge is high or strong enough to overtop, overflow or breach the

dykes, a coastal flood occurs. The study only considers surge induced coastal flooding due to dyke overtopping and overflows.

**Urban environment**: The dyke system is implicitly included in the hydrodynamic processes to set up the boundary conditions of overflow and overtopping discharge that causes coastal flooding. The conceptualization does not include any other infrastructure.

The flood model in this study is based on extreme storm surge scenarios and two-dimensional (2D) hydrodynamic models

explained in (Naulin et al., 2012; Ujeyl and Rose, 2015). The storm surge is composed of wind surge, local tides and a possible external surge due to cyclones. The extreme storm surge events are computed by considering the highest observed occurrence of each component. The three storm surge events – Event A, Event B and Event C – used in this study has a peak water level of 8.00 m, 7.25 m and 8.64 m, respectively (Naulin et al., 2012). Numerical 2D hydrodynamic models are used to calculate water levels and wave stages around the dyke ring. In turn, these data are used to compute the overflow and wave overtopping

discharges for the three scenarios.



To assess the flood hazard from the three scenario events, flood models that simulate coastal flooding are implemented. The model is developed using the MIKE21 unstructured grid modelling software (DHI, 2017). The 2D model domain defines the computational mesh and bathymetry, in which the latter is based on a digital terrain model (see Figure 7). The surface resistance is expressed using a space-dependent Manning number that is based on the current land use categories. The time-dependent

overflow and overtopping discharges over the dykes described above are used as boundary conditions. The output of the hydrodynamic model relevant for the current study is the inundation map showing the maximum flood depth in Wilhelmsburg. This is because the main factor that significantly contributes to building and contents damage is the flood depth (Kreibich and Thieken, 2009). Further, as houses are represented by polygon features (see Figure 1), the flood depth for a specific house is the maximum of the depths extracted for each vertex of the polygon that defines the house.

**4.3 Coupled model factors and setup**

The input factors of the coupled ABM-flood model are presented in Table 2. The input factors are grouped into two. The first group includes the initial conditions and parameters that are regarded as control variables. Varying these factors is not of interest for the study; and hence, they are not included in the model experimentation. That said, a sensitivity analysis (SA) is carried out on these factors to assess which of them contribute more to the uncertainty of the model output. A detailed

discussion of the SA we carried out is given in Appendix D. The second group comprises of factors that are used to set up model experimentation and to evaluate the effect of household adaptation measures in FRM. In this group, the first three factors are related to the shared strategies defined in Table 1, while the last three are related to individual strategies. The flood event scenario is a randomly generated storm surge events series (see Figure 8). The percentage base values in Table 2 are respective to the total number of agents.

Due to the lack of available data, most of the factors are parametrised based on our expert estimations. Some, however, are based on literature or census data. For example, since the last major flood occurred in 1962 and only 14% of Wilhelmsburg's residents are older than the age of 65 (according to the 2011 census[1]), the FE attribute of 86% of the agents is randomly initialized as *No*. The climate change-related thresholds (see also **Eq 3**) are based on a study on country level concern about climate change in which 44% Germans are "very or extremely worried", 42% are "somewhat worried" and the remaining 14%

are "not at all or not very worried, or does not think climate change is happening" (NatCen Social Research, 2017). However, the study does not directly relate climate change with flooding. According to the 2011 census, in Wilhelmsburg, the share of apartments occupied by the owners was 15% while apartments rented for a residential purpose were 82%. The remaining 3% were vacant. Based on that, in the ABM model, we randomly initialise 15% of the households as owners of the houses they occupy while the remaining 85% as renters, assuming that the 3% vacant apartments can potentially be rented. Finally, since

---

[1] Interactive maps for Hamburg for the 2011 Census can be found at https://www.statistik-nord.de/fileadmin/maps/zensus2011_hh/index.html



income is considered sensitive information, the data is not readily available. Hence, we randomly initialise 30% of the agents
as low-income households and the rest as high-income.

The response factors we use to measure the model outcome are the cumulative number of household agents that positively appraised coping ($Coping_{Yes}$), that positively appraised coping due to the social network element ($Coping_{Yes,SN}$), that implemented primary measures ($PM_{implemented}$), that abandoned primary measures ($PM_{abandoned}$), that implemented

secondary measures ($SM_{implemented}$) and that abandoned secondary measures ($SM_{abandoned}$). In terms of damage, we focus on the building and contents damage mitigated rather than the total damage to highlight the benefits of household adaptation measures.

### 4.4 Model verification and validation

As mentioned in Section **Error! Reference source not found.**, the flood model we utilize in this study was developed and

reported in a previous publication. Hence, we take the calibration and validation of the flood model at face value. Regarding the ABM, we carried out model verification by evaluating the relationship between agents' actions and expected response factors. For example, when agents implement measures, system-level number of secondary measures implemented cannot be higher than the primary measures implemented. Or, in coping appraisal, with an increase in the number of agents with high income, we expect a system-level increase in the number of coping agents. However, the average number of agents that

implement permanent measures should not be influenced as there is no relationship between income and permanent measures implementation as specified in the conceptual model.

Regarding the model validation, we validated the conceptual model using expert and local knowledge of the study area. Currently, there is no practice of implementing household adaptation measures in Wilhelmsburg. The study is looking into the potential future direction of reducing vulnerability using a "what-if" approach. Thus, due to the modelling approach performed,

undertaking classical validation is not possible. Given the limitations, the practical purpose of the ABM is to showcase the benefits of household adaptation measures so that authorities and communities in Wilhelmsburg may consider implementing such measures to mitigate potential damages. Moreover, the model serves the purpose of advancing scientific understanding and modelling of socio-hydrologic systems, particularly human-flood interactions.

### 4.5 Experimental setup

To evaluate the effect of the shared strategies listed in Table 1 and individual strategies such as delaying the implementation of measures, implementing secondary measures and abandoning measures, we set up simulations by varying the values of selected input factors as presented in Table 3. The subsidy levers 1, 2 and 3 represent no subsidy, subsidy only for flooded household agents and subsidy for all agents that consider flood as a threat, respectively. Considering the computational cost of simulations, we evaluate six flood event scenarios. The event series of the scenarios are randomly generated and shown in

Figure 8. In these batch of simulations, all the other input factors are set to their base values, as stated in Table 2.





The simulation period of the ABM is 50 time steps in which each time step represents a year. The number of household agents is 7859. Every simulation of parameter combinations is replicated 3000 times. Hence, for the SA and policy-related experiments, simulation outputs are computed as averages of 3000 simulations per input factor setting. A detailed description of estimating the simulation replication is provided in Appendix C. All simulations in this study are performed using the

SURFsara high performance computing cloud facility (https://userinfo.surfsara.nl/systems/hpc-cloud).

## 5 Results

### 5.1 Effects of flood event scenarios

We have tested six different flood event scenarios, and the adaptation behaviours of agents are shown in Figure 9. The plots show that each scenario results in a unique trajectory of adaptation measures. However, Scenarios 1, 3, 4 and 6 have similar

curves of $PM_{implemented}$ while Scenarios 1 and 4 appear to overlap. Irrespective of the subsidy lever, the four scenarios have a similar number of $PM_{implemented}$ at the end of the simulation period. In these scenarios, the biggest event (Event C) occur as the first or the second event. As this event is big enough to flood every agent's house directly, most agents tend to develop protection motivation behaviour earlier. On the other hand, Scenarios 2 and 5 display a lower number of the response factor, which improves with a subsidy. In these scenarios, Event C occurs last; and hence, the $PM_{implemented}$ rises rapidly after

$time\ step = 35$.

In terms of building damage mitigated, the scenarios with the two big events (C and A) occurring as first and second and within a short time interval display the least damage mitigated (see Figure 9 (b) Scenarios 4 and 6). These are considered to be the worst cases of the six scenarios as agents did not have a coping behaviour before the first big event, and most agents did not yet develop coping behaviour when the second big event occurred after five years. Only 21% and 14% of the agents

implemented a measure in cases of Scenarios 4 and 6, respectively, without subsidy. In contrast, in the case of Scenario 5, agents gradually develop coping behaviour after a first big event. By the time the second big event occurred after 37 years, about 45% and 70% of the agents already implemented a primary adaptation measure without subsidy and with a subsidy to flooded houses, respectively. Scenario 5 can be considered as the best scenario in which household agents have time to adapt and significantly reduce the potential damage that may occur in the future.

The main lesson from the results of the scenario exercise is that agents should be prepared or adapt quickly after an event to mitigate considerable potential damages. Big events may occur within a short time interval, and households should be prepared to mitigate associated damages. It should be noted that in Figure 9 (b) there is no mitigated damage in the first event as we assumed that no mitigation measure was implemented initially.





## 5.2 Impacts of subsidies and shared strategies

The effects of the institutions are analysed in two categories. The first ones are the impacts of subsidies, and the second effects are that of the social network and shared strategy parameters.

**Impacts of subsidies**: The cumulative number of implemented primary measures plotted in Figure 10 shows that providing subsidies increases the protection motivation behaviour of agents irrespective of the flood event scenarios. For example, in the case of Scenario 1 flood event series, the building damage mitigated increases by about 130% when a subsidy is provided to

agents (see Figure 9 (b)). However, giving subsidies either only to flooded agents or to all agents does not have a difference in the coping responses of agents. That is depicted by the overlapping curves of $SS = 2$ and $SS = 3$ in Figure 10. The result can be justified by the fact that (i) the subsidies only affect agents that implement permanent measures; and (ii) when a big flood event happens, it floods most of the agents, essentially levelling the number of agents impacted by $SS = 2$ and $SS = 3$.

**Impacts of social network and shared strategy parameters**: Figure 11 shows that an increase in the value of the social

network parameter reduces the number of agents that develop a coping behaviour. As the $SN$ parameter is associated with the proportion of coping agents within a house category, a higher $SN$ requires a majority of agents in a given house category should have developed a coping behaviour to start influencing other agents. For example, when $SN = 0.5$, no agent is influenced by their social network as the criteria that at least 50% of the agents in the same house category should have already implemented a measure to influence others has never been satisfied. On the other hand, when $SN = 0.2$, about 75% of the

agents that developed a coping behaviour after $time\ step = 20$ are influenced by their social network. Figure 11 also shows that the shared strategy parameter does not have a significant effect on the number of agents that develop a coping behaviour (for example, see the solid lines cluster together). This means that when the SN criteria are satisfied, most agents tend to follow the crowd.

In practical terms, this result shows that if agents need to wait to see many others implement measures to be influenced, most

likely, they will not develop a motivation protection behaviour. Hence, aspects such as stronger community togetherness in which few neighbours can influence others to increase the possibility of implementing adaptation measures.

## 5.3 Impacts of individual strategies

In this section, we will analyse the effects of three factors that characterise individual strategies: delay parameter, adaptation duration and secondary measure parameter.

**Impacts of delay parameter**: As shown in Figure 12, the percentage of agents that transform the coping behaviour to action decreases as the value of the delay parameter increases. When $DP = 1$, all agents that developed coping behaviour implement adaptation measures at the same time step. However, when $DP = 9$ (i.e., when the probability that a coping agent will implement a measure at a given year is 1/9), the number of agents that implement measures is 75% of the number that develop a coping behaviour by the end of the simulation period.





Furthermore, both the number of coping agents and agents that implemented measures decreases with increase in $DP$ value. For example, when $FE_{scenario} = 2$ and the value of $DP$ increases from 1 to 9, the numbers of coping agents and agents that implemented a primary measure drop by about 27% and 48%, respectively, at $time\ step = 50$. This also has a knock-on effect on the implementation of a secondary measure, which reduces by about 50%. Based on the outputs of the simulations, the delayed implementation of measures reduces the potential building and contents damage that could have been mitigated by

€36.3 million and €8.7 million, respectively.

The main reason for the lower number of measures implemented with the increase in the value of the delay parameter is the decision of agents to delay the implementation. However, that also contributes to lower the number of agents influenced by their social network. In practical terms, this means that authorities should support households who tend to develop protection motivation behaviour so that they would implement adaptation measures promptly.

**Impacts of adaptation duration parameter**: We evaluate the impacts of the adaptation duration using the number of agents that implemented and abandoned primary and secondary measures. The simulation results in Figure 13 (a) show that the adaptation duration parameter has a minor impact on the number of primary and secondary measures implemented, regardless of the subsidy lever. For example, the largest percentage difference between the highest and lowest $PM_{implemented}$ is exhibited around $time\ step = 30$, which accounts about 28%. One reason for the minor impact of $Y_{adaptation}$ on $PM_{implemented}$ could

be that the parameter only affects agents that implement temporary primary measures, which is about half of the total number of agents. Another one could be that an increase in $PM_{implemented}$ also increases the number of agents that potentially abandon the measure. This is reflected in Figure 13 (b) in which the peaks of $PM_{abandoned}$ correspond to the steepest slope of the curve displaying $PM_{implemented}$.

Figure 13 (b) also shows that more agents abandon measures when the value of $Y_{adaptation}$ decreases. But then the number of

measures abandoned decreases as agents reach the fixed number of times they could abandon measures, which is specified by the $f_{abandoning}$ parameter. In addition, the figure illustrates that, in general, $SM_{abandoned}$ is larger than $PM_{abandoned}$ along the simulation period. This can be explained by the model conceptualization, where agents first abandon secondary measures provided that they consider implementing them.

The practical lesson from the simulation results is that if agents tend to implement temporary measures, there should be a

mechanism that encourages them to continue implementing the measures in future. For example, authorities may create and raise public awareness of how to seal windows and doors, and the availability of sandbags. This should be done regularly, and especially just before the event occurs as the measures can be implemented within a short period.

**Impacts of secondary measure parameter**: Finally, we analyse the impacts of $SMP$ on the number of agents that implemented secondary measures. Since the secondary measure conceptualized in the model is adapted furnishing, the effects

of $SMP$ are evaluated based on the contents damage mitigated.

Figure 14 (a) shows that the cumulative number of agents that implemented secondary measure increases as the parameter value increases. But, the rate of increase in $SM_{implemented}$ is marginal especially for $SMP \geq 0.4$, in both cases of subsidy





levers. When flooded agents receive a subsidy, $SM_{implemented}$ increases by about 1000 agents compared to the policy lever with no subsidy. Although the subsidy does not directly affect the implementation of secondary measures, it increases the implementation of primary measures, which in turn, increases $SM_{implemented}$. The only exception is when $SMP = 0$; in that case, no agent implement secondary measure despite the subsidy lever.

Similarly, Figure 14 (b) shows that the contents damage mitigated increases marginally with the increase in the $SMP$ value. The damage mitigated when $SMP = 0$ is because some agents implemented flood adapted interior fittings, which are classified as primary measures, and these measures mitigate both building and contents damages. When there is a subsidy, the contents damage mitigated increases by about three folds for each of the $SMP$ values, except $SMP = 0$, compared to the policy lever with no subsidy.

The marginal increases in the $SM_{implemented}$ and the contents damage mitigated together with the increase in the values of SMP is because not all agents could implement secondary measures. As discussed in the model conceptualisation, agents that live in bungalows and garden houses do not implement adapted furnishing since those house categories are single-storey houses. In general, based on our simulation outputs, implementing only a secondary measure could mitigate more than €40 million. Hence, decision-makers should encourage households to consider implementing such simple measures that could be done at no monetary cost provided that there is space to keep contents safe.

## 6 Discussion and conclusion

The study aims to improve the current modelling practices of human-flood interaction and draw new insights for FRM policy design. Below, we discuss our modelling contributions and how they leads to policy insights.

i.  We have incorporated occurrences of flood events to examine how that influence household agents' adaptation behaviour. In our study, we examined six flood event scenarios, each comprising of three coastal flood events occurring within 50 years simulation period. Simulation results show that a unique trajectory of adaptation measures and flood damages emerge from each flood event series. The interval between the occurrences of two big events is an important factor in defining households' adaptation behaviour. If a big event occurs first, it can serve as a wake-up call for future coping behaviours. However, that comes with a substantial amount of building and contents damage. Households and authorities in Wilhelmsburg should avoid maladaptive practices (in PMT terms) such as avoidance and denial of possible future flooding and implement a measure to mitigate potential damages.

ii.  We have analysed the effects of a subsidy on the adaptation behaviour of individuals. We tested three subsidy levers: no subsidy, subsidy only for flooded household agents and subsidy for all agents that consider flood as a threat. Based on the simulation results, the last two levers have similar outcomes in terms of coping behaviours. It may depend on the flood event series, but providing subsidies increases the number of coping households in the long run. Hence authorities in Wilhelmsburg may consider providing subsidies to motivate households that implement permanent measures.

iii.  We have formulated the implementation of adaptation measures as informal institutions in the form of shared strategies that are influenced by social networks. Simulation results reveal that a wait-and-see approach, such as a high social network parameter settings, does not help to increase the number of coping households. There should be an approach in which fewer group of trusted community members or public figures may influence others in their community.



iv. We have also analysed the effect of individual strategies on household adaptation behaviour. The strategies are delaying the implementation of measures, decisions on the adaptation durations of temporary measures and implementing
secondary measures. Simulation results show that delaying measures implementation reduces millions of euros that could have been mitigated. On the other hand, the overall impact of longer adaptation duration by some households could be cancelled out by the decision to abandon measures by others. It is essential to raise awareness continuously so that households do not forget or abandon to implement temporary measures. The role of simple measures such as adapted furnishing, which do not incur any monetary cost, should also be highlighted as these measures could contribute to
reducing millions of euros of contents damages.

In conclusion, the paper presented a coupled agent-based (ABM) and flood models developed to evaluate the adaptation behaviour and decision making of households to implement vulnerability reduction measures in the Wilhelmsburg quarter of Hamburg, Germany. We have employed the coupled flood-agent-institution modelling (CLAIM) framework to conceptualize
the agent-flood interaction in the coupled model, and the protection motivation theory (PMT) to study household flood preparedness behaviour. The model conceptualization has benefitted from the qualitative exploration of PMT carried out in the same study area. Adding local knowledge of flood risk management (FRM) issues and using other data sources, we extended the previous work by developing a simulation model that could support decision-making. Furthermore, the study has extended other prior works (Abebe et al., 2019b; Erdlenbruch and Bonté, 2018; Haer et al., 2016) to study human-flood
interaction better and to gain new policy insights. With all the extensions, we have demonstrated that coupled ABM and flood models, together with a behavioural model, can potentially be used as decision support tools to examine the role of household adaptation measures in FRM. Although the focus of the paper is the case of Wilhelmsburg, the improved modelling approach can be applied to any case to test policy levers and strategies considering heterogeneous individual behaviours.

The model conceptualization and the results would benefit from further refinement to provide more accurate insights into
policy design. For example, more representative datasets are needed to reduce the input factors uncertainty as indicated by the sensitivity analysis (see Appendix D). In our model conceptualization, households implement specific measures based on the category of a house they occupy, as defined in the shared strategies. Those are expert-based hypothetical strategies that could have been defined otherwise. Thus, the modelling exercises and their outcomes should be seen as an effort (i) to advance the use of coupled ABM-flood models in FRM, and (ii) to provoke communities and decision-makers in Wilhelmsburg to
investigate further the role of household adaptation measures in mitigating potential damages. Furthermore, it is important to note that while the existing work addressed household measures, the same approach can be also applied to a range of different measures and contexts (e.g., local and regional measures, nature-based solutions and traditional "grey infrastructure") which we intend to address in our future work.

Finally, the research presented can be enhanced by analysing model uncertainty. One may conceptualise the ABM differently,
and investigating the impact of the different model conceptualization would be essential to communicate the uncertainty in model results. The research objective could also be extended by including other types of agents such as businesses and industries, and other response factors such as indirect damages (e.g., lost revenues due to business interruptions) to provide a broader view of the role of individual adaptation measures.





**Appendix A – List of assumptions made to build the coupled ABM-flood model**

To structure and conceptualize the Sint Maarten flood risk management case and develop the agent-based model, we have made the following assumptions. The reasons to make these assumptions are model simplification (i.e., to develop a less complicated model) and lack of data.

1. Household agents are spatially represented by the houses they live in; hence, they are static.

2. There is a one-to-one relationship between household agents and houses (i.e., a household owns only one house and vice 620 versa).

3. Houses are represented by polygon features such that each polygon represents one household agent. In the case of multi-storey buildings, the agent represents the household(s) living on the ground floor.

4. When apartments and high-rise buildings are represented by one single polygon feature, the whole building is considered as one house representing one household agent.

5. A maximum of one flood event occurs per time step.

6. Only three flood event scenarios are considered. All the scenarios simulate dyke overtopping and have very low exceedance probability. Dike breach is not considered in the conceptualization.

7. When there is flood, the flood depth of a house is extracted from the flood maps as the maximum of the flood depths read at the vertices of the polygon feature that represent the house.

8. A house is considered to be flooded if the flood depth is greater than 10 cm assuming that all houses have floor elevation of at least 10 cm.

9. Damage assessment does not include aspects such as damages on other assets (e.g., cars), indirect damage (e.g. business interruptions), risk to life, and structural collapse of buildings.

10. Damage is assessed based only on the flood water level. The effect of floodwater velocity, duration and contamination 635 level is not included in the damage assessment.

11. Both building and content damages are assessed per building type. The damages of all houses of the same building type are calculated using the depth-damage curves for that building type.

12. The sources of information does not initiate the coping appraisal process as in the original PMT study as agents know the kind of measure they implement.

13. If a house has already appraised coping and implemented a measure, they don't appraise coping again, unless they abandon the measure, assuming that they do not implement another primary measure.

14. Adaptation measures are sufficient to reduce flood damage in all flood events (perceived efficacy of measures).

15. Agents are capable of successfully implementing adaptation measures (perceived self-efficacy).

16. The effect of flood barriers such as flood protection walls and sandbags on the flood hydraulics is not accounted.

17. Agents only implement a maximum of one primary and one secondary measure at a given time step.





18. Agents do not implement temporary adaptation measures (i.e., flood barriers) at any time step but deciding to implement the measures entails they only deploy them when there is flood.

19. If agents abandon measures, they only abandon non-permanent measures such as flood barriers.

20. In case of non-permanent measures, if a household agent decides to implement a measure, the decision is valid at least for a year.

21. If a household agent abandons a measure, it abandons it for at least a year.

22. Household agents do not implement the same primary measure twice unless they abandon it.

23. The adaptation duration specified in a simulation is the same for all temporary measures.

**Appendix B – List of house types in Wilhelmsburg**

EFH30A – Single-family house, Thermal insulation composite system

EFH30B – Single-family house, Cavity wall with insulation

EFH31A – Single-family house, plastered brick work, ground level: raised ground floor

EFH31B – Single-family house, plastered brick work, Souterrain/basement

EFH32A – Single-family house, plastered brick work

EFH32B – Single-family house, faced brick work

EFH34 – Single -family house, plastered brick work, Souterrain: apartment

EFH35A – Bungalow, plastered brick work

EFH35B – Bungalow, wooden construction

KGV33A – garden/summer house, plastered brick work

KGV33B – garden/summer house, wooden construction

MFH20A – Apartment building, basement: water proof concrete tanking

MFH21A – Apartment building, plastered brick work, ground level: apartments

MFH21B – Apartment building, faced brick work, ground level: apartments

MFH21C – Apartment building, faced reinforced concrete, ground level: apartments

MFH22A – Apartment building, faced brick work, ground level: business

MFH22B – Apartment building, faced brick work, ground level: business (same as MFH_22a)

MFH23A – Apartment building, plastered brick work, ground level: apartments

MFH23B – Apartment building, faced brick work, ground level: apartments

MFHH10 – High-rise building, dry construction, ground level: general use

MFHH11 – High-rise building, reinforced concrete, ground level: general use

MFHH12 – High-rise building, dry construction, ground level with garages

IGS – Hybrid house – IGS centre





OH – Hybrid house – Open house

HH – Hybrid house

SIG – Phase change material – smart is green

BIQ – Smart material house – BIQ

CS1 – Smart price house

GUS – Smart price house – Grundbau und Siedler (Do-it-yourself builders)

WH – Wälderhaus

WC – Wood Cube

**Appendix C – Estimating simulations repetition**

ABMs are often stochastic. For example, agent behaviours are determined based on random values generated from pseudo-random numbers, which produces results that show variability even for the same input factor setting (Bruch and Atwell, 2015; Lorscheid et al., 2012; Nikolic et al., 2013, p.110–111). Hence, reliable ABM outputs are obtained by running simulations

multiple times. To determine the number of simulation runs, we apply the experimental error variance analysis suggested by Lorscheid et al. (2012). The coefficient of variation ($c_v$) is used to measure the variability in the model output. Starting from a relatively low number of runs, the $c_v$ of the model output is calculated by increasing the number of runs iteratively for the same factor settings. The number of runs is fixed when the $c_v$ stabilizes or the difference between the $c_v$'s of iterations falls below a criterion. This experiment is done for selected input factor settings to cross check whether output variations stabilize

around the same number of runs irrespective of the factor settings. We evaluate the $c_v$'s for the six response factors.

We iteratively run simulations starting from 100 to 5000 and compute the $c_v$'s of six response factors for each iteration, for several input factor settings. As an example, Table C-1 shows the $c_v$'s for the factor setting in which all the input factors have the base values. Selecting a difference criterion of 0.001, the minimum sample size in which the $c_v$'s start to stabilize is 3000. As the $c_v$'s do not change while increasing the number of runs, we fix the number of runs to be 3000. For the SA and policy-

related experiments, simulation outputs are computed as averages of 3000 simulations per input factor setting.

**Appendix D – Sensitivity analysis**

As in any model, the ABM developed in this study is subject to uncertainties. Regarding input factors uncertainty, the initial conditions and parameters mentioned in Table 2 are either based on our expert estimations or based on available coarse datasets such as the 2011 national census in Germany. Hence, a sensitivity analysis (SA) is carried out to allocate the model output

uncertainty to the model input uncertainty. The SA method adopted in this study is the *elementary effects (EE) method*, also called the Morris method (Morris, 1991). The method is effective in identifying the important input factors with a relatively small number of sample points (Saltelli et al., 2008, p.109). Saltelli et al. explained that "the method is convenient when the



number of factors is large [and] the model execution time is such that the computational cost of more sophisticated techniques is excessive" (p. 127). We employ this method because of the high computational cost related to the large number of simulation
repetitions estimated (see Appendix C).

The EE method is a specialized one-at-a-time SA design that removes the dependence on a single sample point by introducing ranges of variations for the inputs and averaging local measures. The sensitivity measures proposed by Morris are the mean ($\mu$) and standard deviation ($\sigma$) of the set of EEs, which are incremental ratios, of each input factor. In a revised Morris method, Campolongo et al. (2007) proposed an additional sensitivity measure, $\mu^*$, which is the estimate of the mean of the distribution
of the absolute values of the EEs. The sampling strategy to estimate the sensitivity measures is building $r$ EE trajectories of $(k + 1)$ points for each $k$ factor, resulting in a total of $r(k + 1)$ sample points. Following (Saltelli et al., 2008, p.119), we choose $r$ to be 10, and each model input is divided into four levels within the input value range. In this study, the input factors selected for the SA are the initial conditions and parameters (as specified in Table 2). Therefore, the computational cost of the SA is $10(10 + 1) = 110$. In Table D-1, we list these factors, their distributions and value ranges. In the SA, the other input
factors presented in Table 2 are set to their base values.

The SA is carried out on the 10 input factors, and the outputs quantify five response factors evaluated at $time\ step = 50$. Figure D-1 shows the Morris sensitivity measures $\mu^*$ and $\sigma$ plotted against each other for five response factors. As $Y_{delay} = 1$ in all the simulations, the response factors $Coping_{Yes}$ and $PM_{implemented}$ have exactly the same value. Hence, only the former response factor is displayed in the figure. The results show that the most important factor by far is $HO_{update}$ though its value
varies only between zero and 2% of the total number of agents. The base value of this factor, representing the change in house ownership, is estimated by the authors of this paper. It is also modelled in such a way that randomly selected household agents may change house ownership state every time step. Considering the influence of $HO_{update}$ on the model output (given the current model conceptualization), it would be essential to acquire reliable data and better model representation of the factor to reduce the model output uncertainties.

The next influential factors are $HI_{update}$, $HO_{ini}$, and $HI_{ini}$. The base values of the household income-related factors are also based on our estimations as there is no publicly available record due to the sensitive nature of income data. Similarly, obtaining better dataset would help to reduce the output uncertainty. The initial house ownership variable is based on census data, but agents' house ownership is assigned randomly as there is no available data regarding its spatial distribution. The $f_{abandoning}$ factor is influential in the case of primary measures abandoning as it sets a limit on the number of times an agent could abandon
a measure. Better data would also reduce this factor's allocation to the model output uncertainty. All the other factors are non-influential as points representing these factors overlap around the (0, 0) coordinate.

**Code availability**

The agent-based model code is available at: https://github.com/yaredo77/Coupled_ABM-Flood_Model_Hamburg



**Author contribution**

YAA, AGo and IN developed the study. YAA, AGo, IN, NM and AGr developed the conceptual model. YAA developed the model code and performed the simulations. YAA prepared the original draft of the manuscript, with further reviewing, and editing from all co-authors. ZV organised funding acquisition.

**Competing interests**

The authors declare that they have no conflict of interest.

**Acknowledgements**

The research leading to these results has received funding from the European Union Seventh Framework Programme (FP7/2007-2013) under Grant agreement n° 603663 for the research project PEARL (Preparing for Extreme And Rare events in coastaL regions), and from the European Union's Horizon 2020 Research and Innovation Programme under Grant agreement No 776866 for the research project RECONECT. The study reflects only the authors' views and the European Union is not
liable for any use that may be made of the information contained herein. We thank SURFsara (https://www.surf.nl/en) for providing a High Performance Computing (HPC) cloud resources that we used to run all the simulations.

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





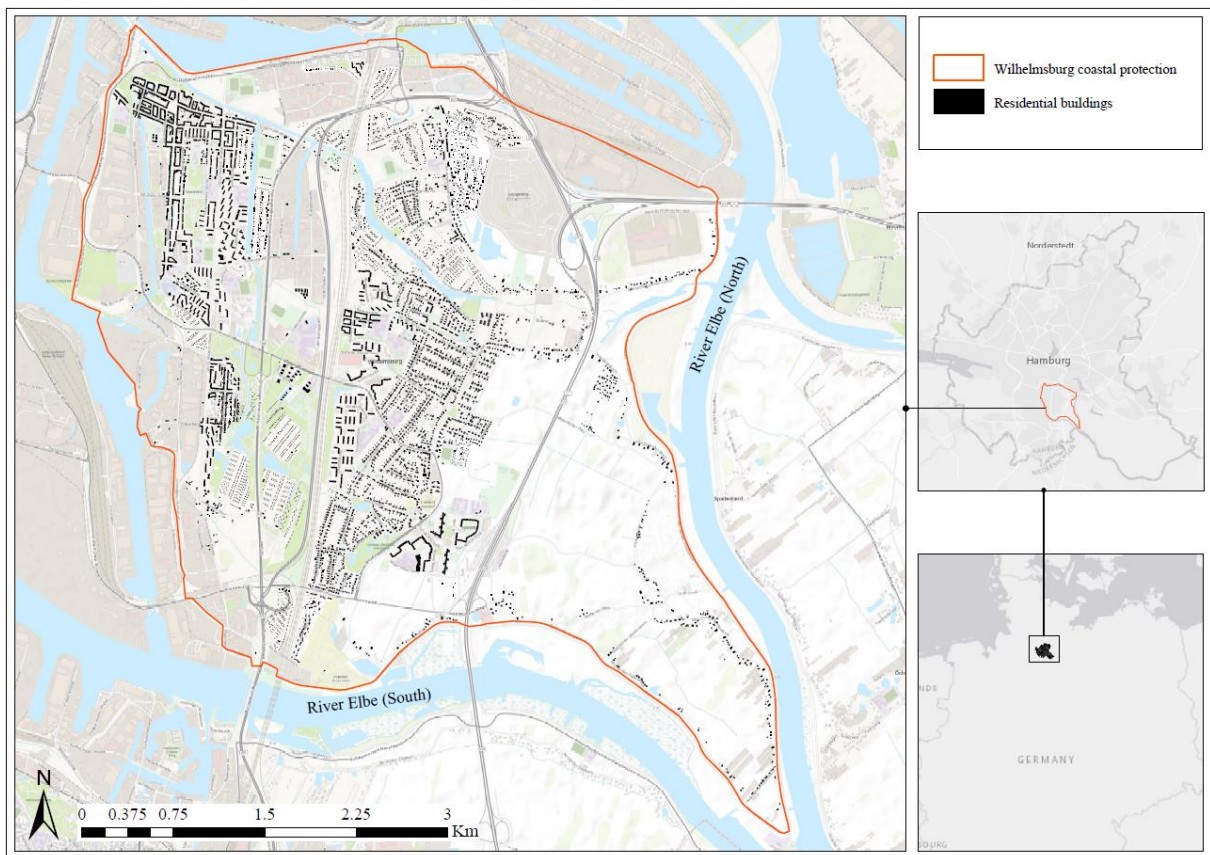

**Figure 1. A map of the study area of Wilhelmsburg. The red polygon shows Wilhelmsburg's coastal protection ring of dykes and walls. The study focuses on residential housings within the protected area. The buildings shown in the map are only those that are part of the model conceptualisation. The inset maps in the right show the map of Germany (bottom) and Hamburg (top). (Source: the base map is an ESRI Topographic Map).**

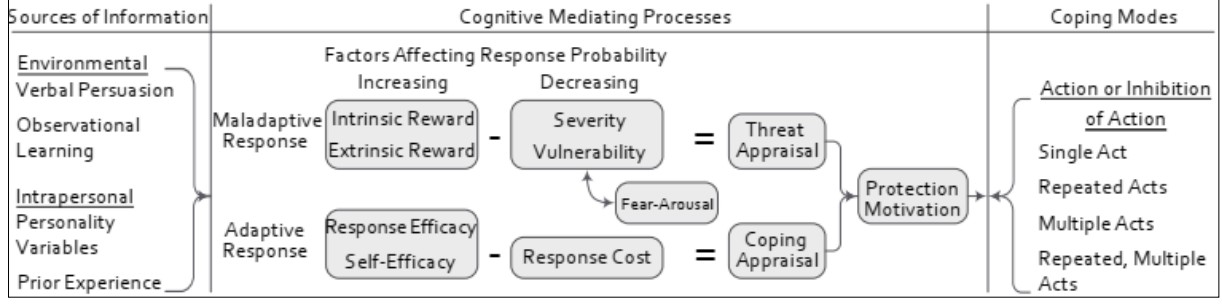

**Figure 2. The original schematization of the protection motivation theory (from Rogers, 1983)**





**Figure 3. CLAIM model implementation flowchart for the FRM case of Wilhelmsburg. (a) shows the general flow chart. (b) shows how implementing individual adaptation measures is modelled in the ABM while (c) shows how measures abandoning is modelled. The rest of the actions shown in sub-process shapes in (a) (shapes with double-struck vertical edges) are shown in figures below. In (b) and (c), RN is a random number, $p_{adaptation,primary}$ and $p_{adaptation,secondary}$ are the probabilities of adapting primary and secondary measures, respectively, and $p_{abandoning}$ is the probability of abandoning a primary or a secondary measure.**


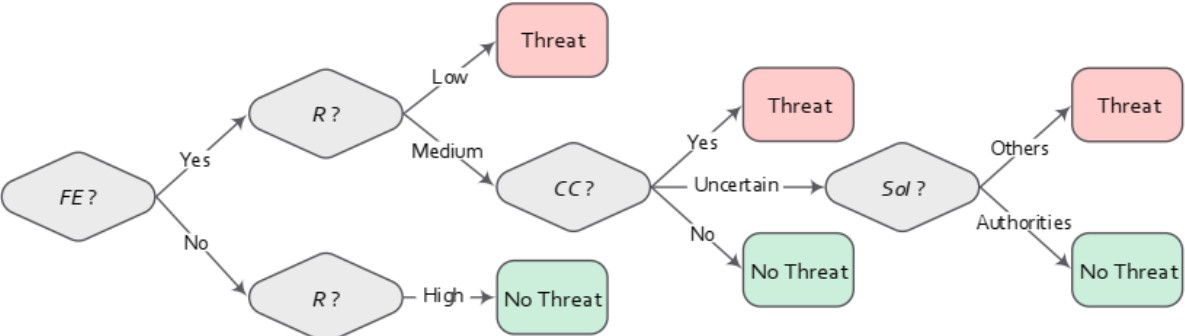

**Figure 4. Decision tree for the threat appraisal**

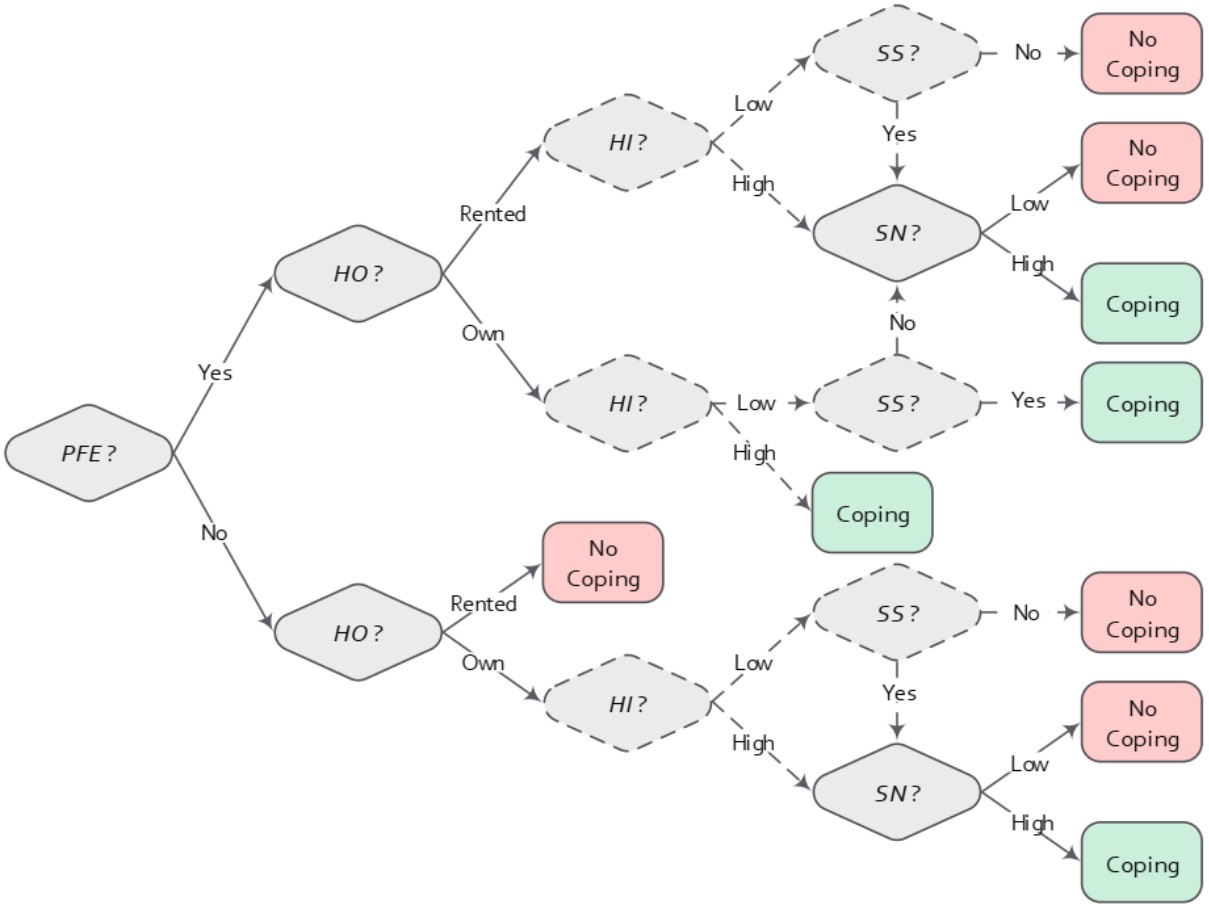





**Figure 5. Decision tree for the coping appraisal. The shapes and lines in dashed line are related to the income and subsidy factors, and they are executed only when households implement structural measures.**

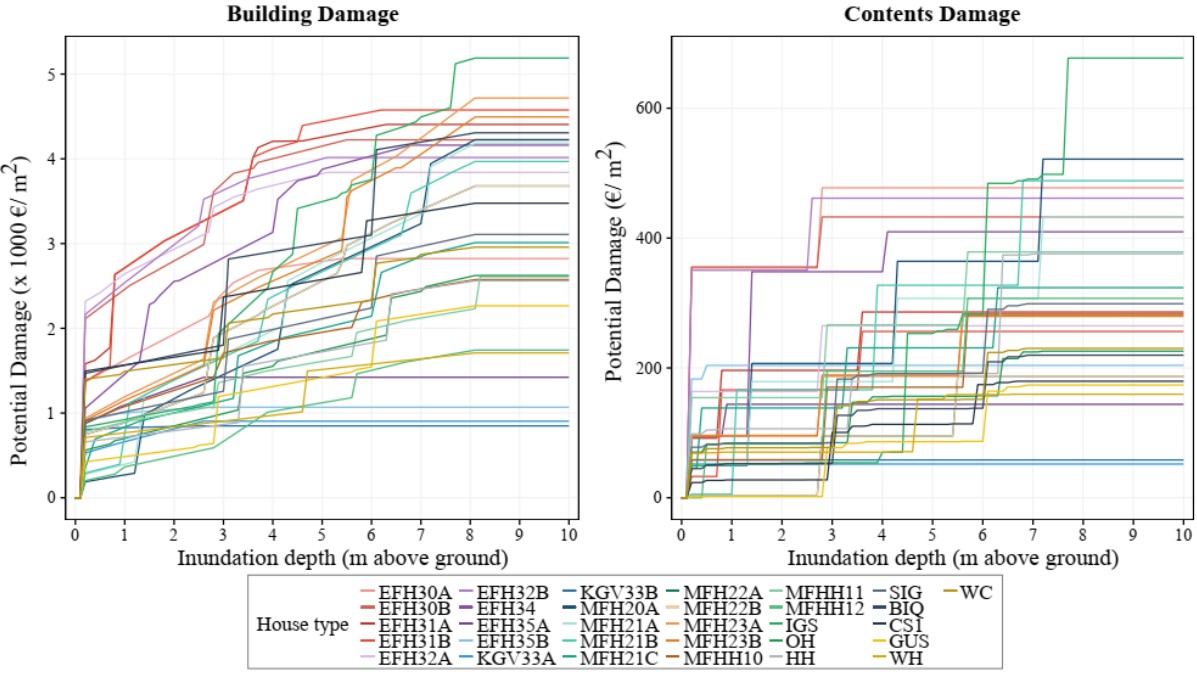

**Figure 6. Depth-damage curves for building (left panel) and contents (right panel) of 31 house types in Wilhelmsburg. A description of the house type codes is given in Appendix B.**

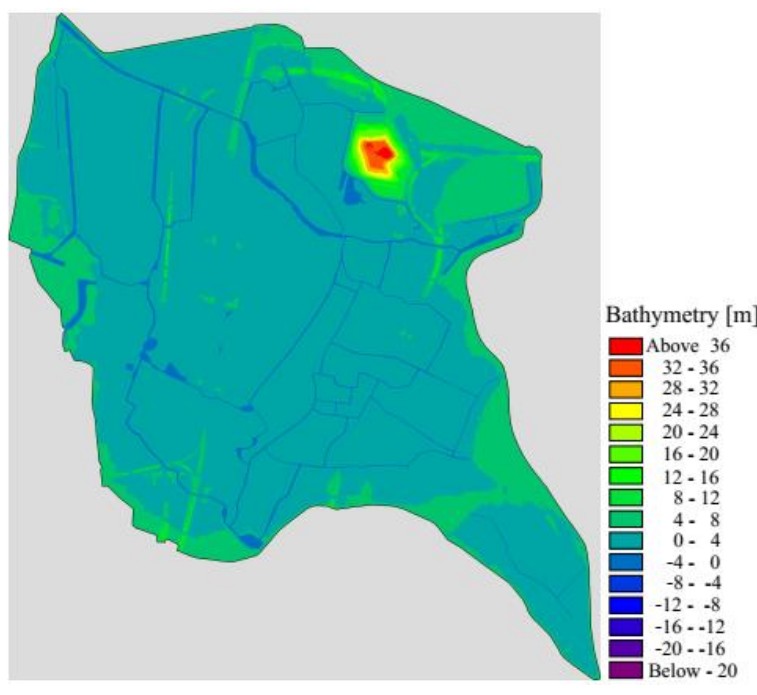






**Figure 7. MIKE21 coastal flood model domain showing the bathymetry**

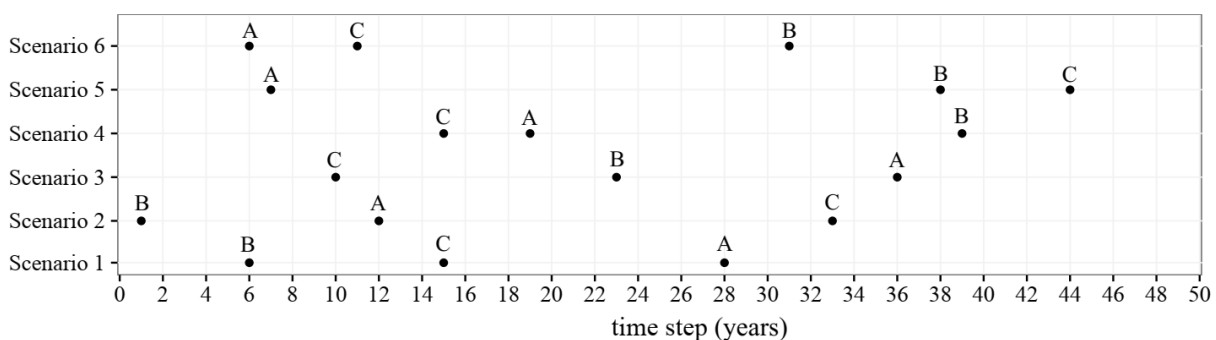

**Figure 8. Scenarios of flood event series. A, B and C represent flood events of storm surge with peak water levels of 8.00 m, 7.25 m and 8.64 m, respectively.**

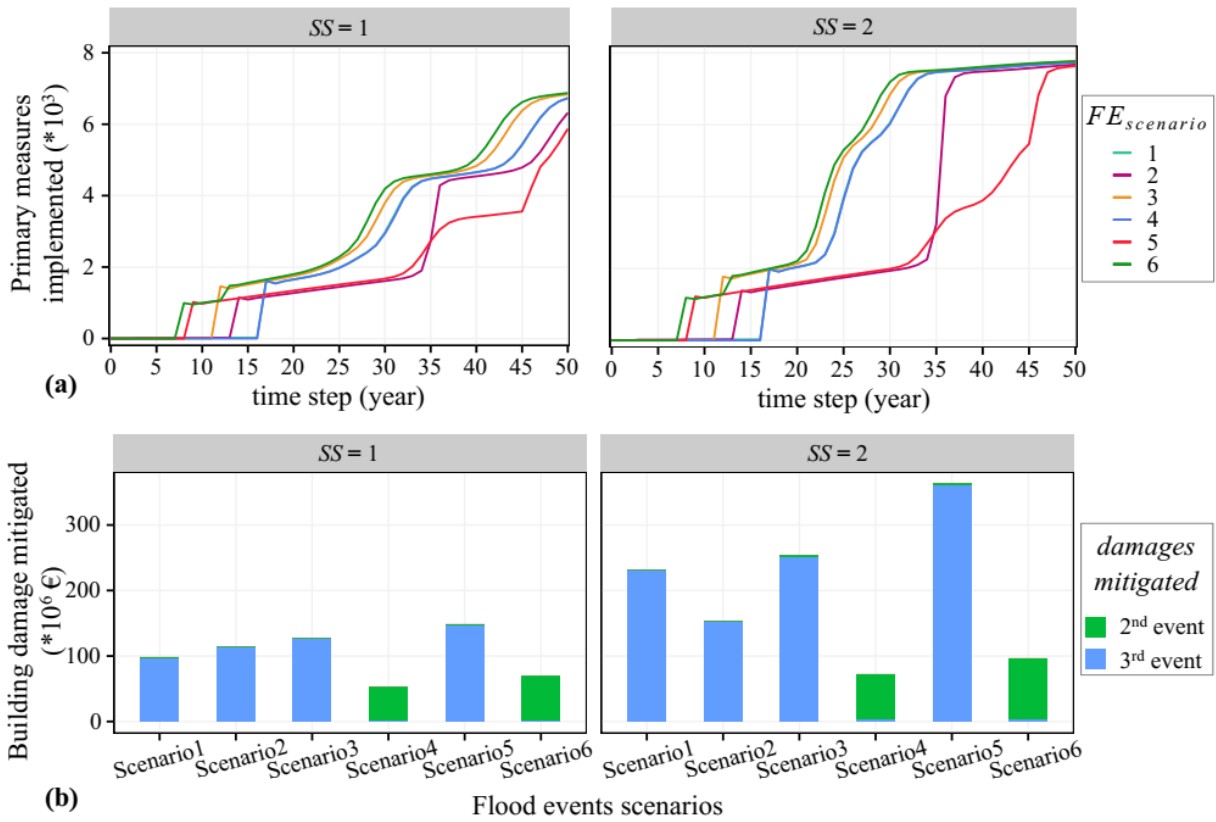

**Figure 9. Effects of six flood event scenarios on the adaptation behaviour of agents and the associated impact. (a) shows the cumulative number of primary measures implemented, and (b) shows the potential building damage mitigated due to the primary measures implemented. In both (a) and (b), the left and right panels show the simulation results without subsidies and with subsidies for flooded agents, respectively.**





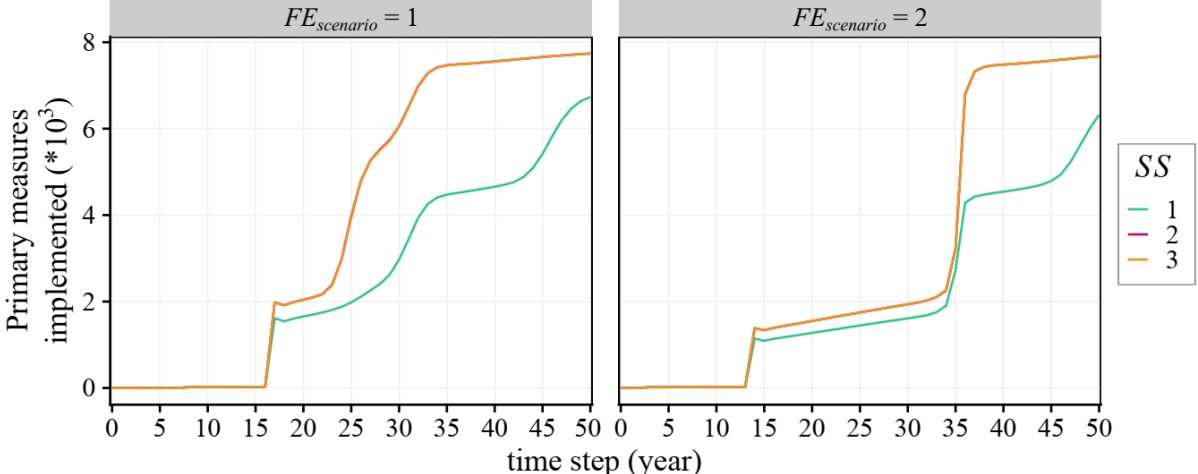

**Figure 10. Impacts of subsidy on the adaptation behaviour of agents. The subsidy levers 1, 2 and 3 represent no subsidy, subsidy only for flooded household agents and subsidy for all agents that consider flood as a threat, respectively. The left and right panels show simulation results with flood events scenarios of 1 and 2, respectively.**

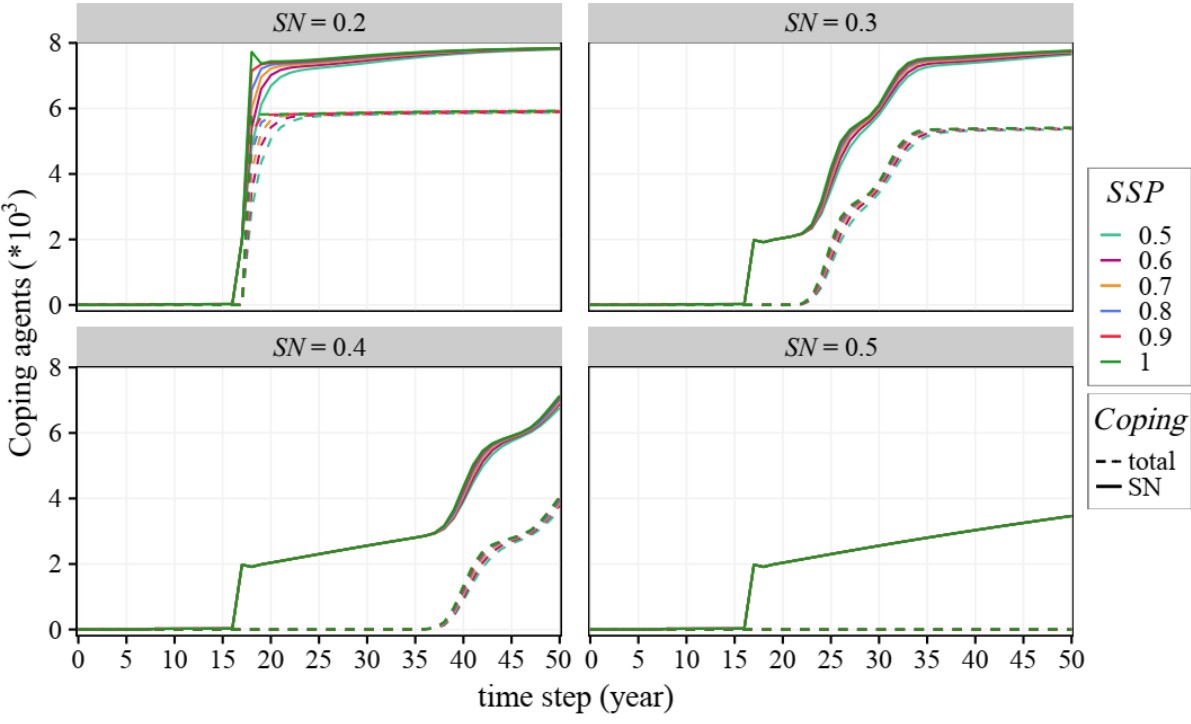


**Figure 11. Impacts of the social network and shared strategy parameter factors on the adaptation behaviour of agents. The solid lines show the total number of coping agents while the dashed lines show the agents that develop a coping behaviour influenced by their social network.**

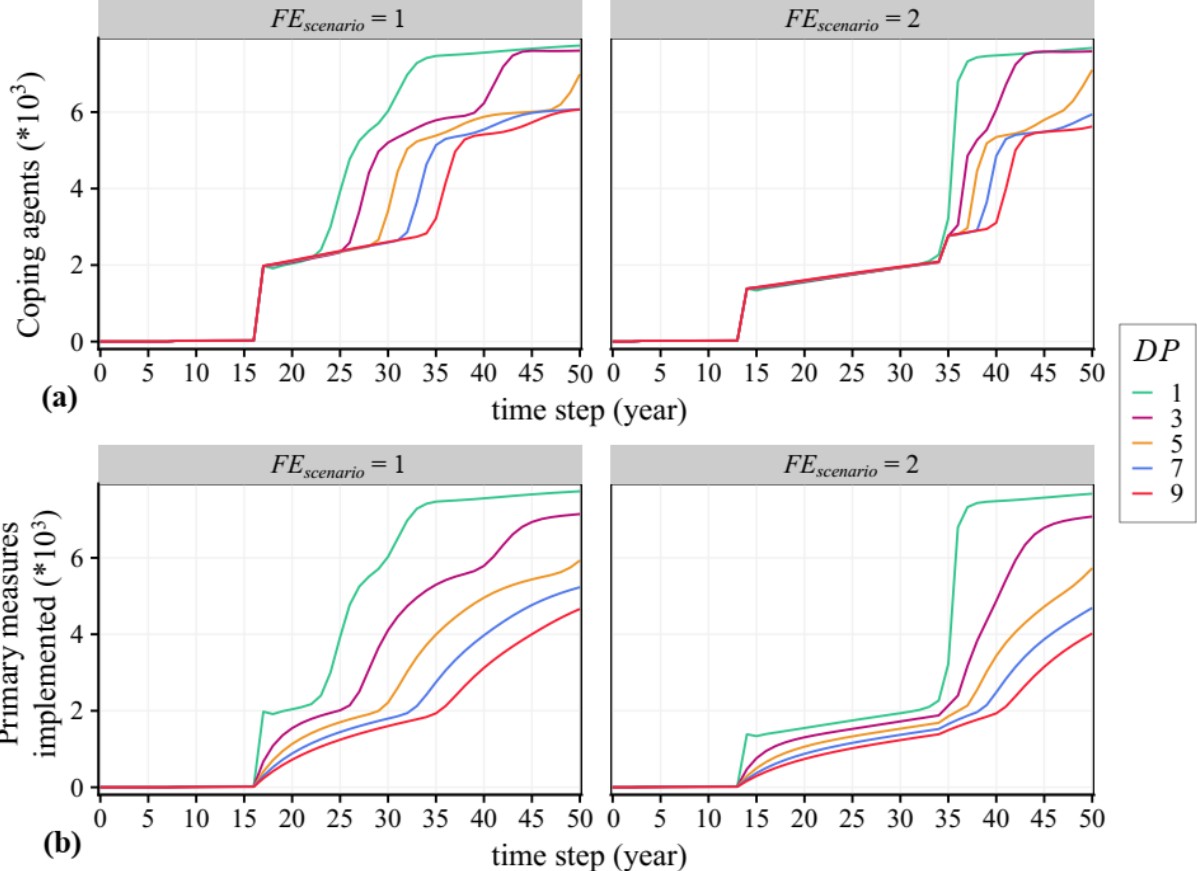

**Figure 12. Impacts of the delay parameter on the adaptation behaviour of agents. (a) shows the coping behaviour of agents and (b) shows the cumulative number of agents that converted their coping behaviour to action, i.e., implement primary adaptation measures. Simulations that generated the results are set with $SS = 2$. The left and right panels show simulation results with flood events scenarios of 1 and 2, respectively.**





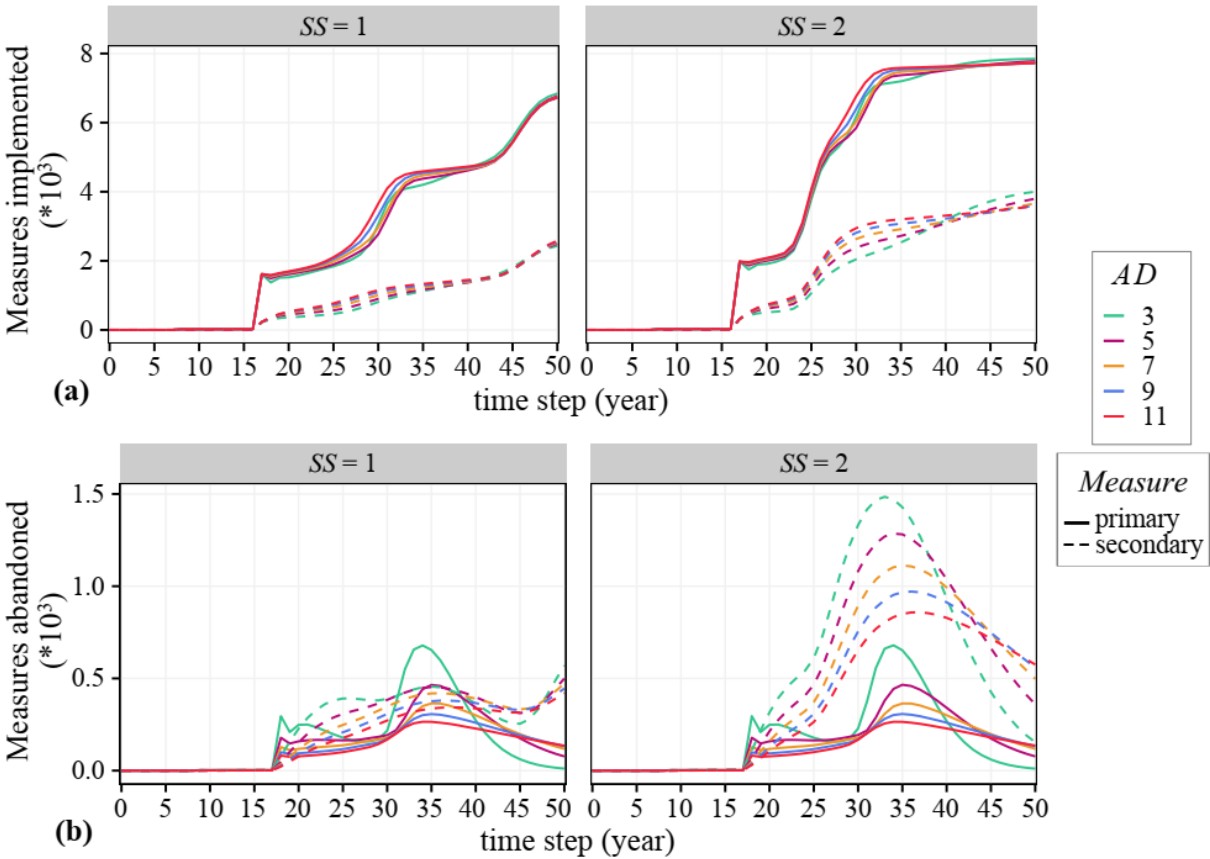

**Figure 13. Impacts of the adaptation duration on the adaptation behaviour of agents. (a) shows the primary and secondary measures implemented, and (b) shows the primary and secondary measures abandoned. The left and right panels show simulation results without subsidies and with subsidies for flooded agents, respectively.**



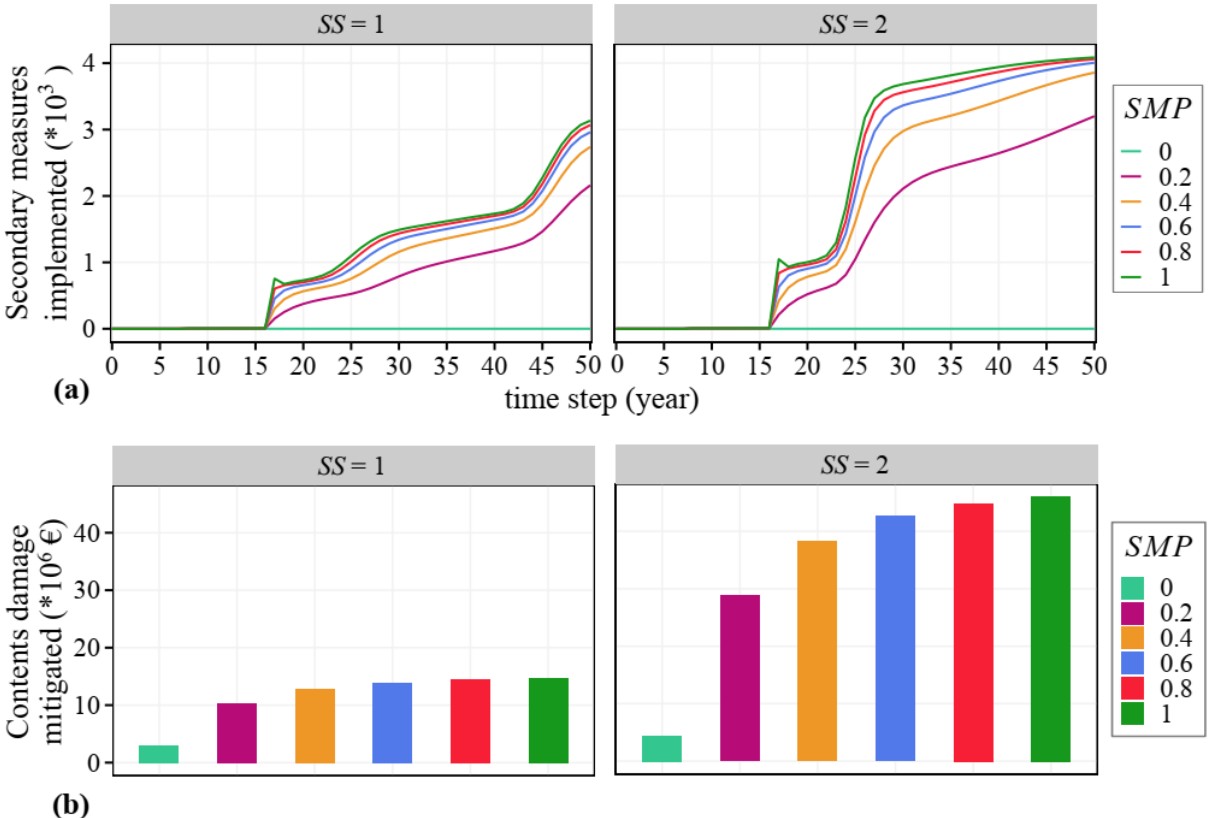

**Figure 14. Impacts of the secondary measure parameter on the adaptation behaviour of agents. (a) shows the cumulative number of secondary measures implemented, and (b) shows the potential contents damage mitigated. The left and right panels show simulation results without subsidies and with subsidies for flooded agents, respectively.**






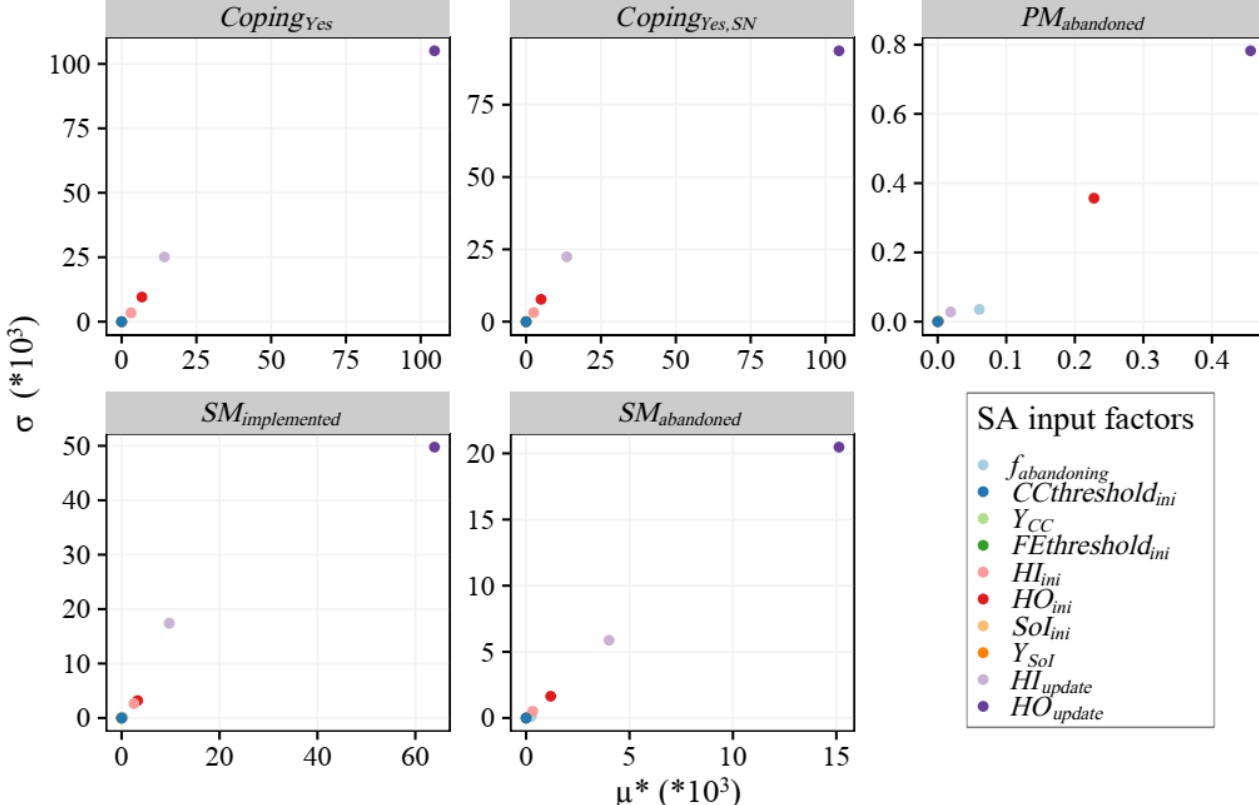

**Figure D-1. Scatter plots displaying the Morris sensitivity measures $\mu^*$ and $\sigma$ for five of the response factors. Points representing the least important factors may not be visible as they overlap close to the (0, 0) coordinate.**


**Table 1. ADICO table of institutions defined for the Wilhelmsburg FRM case.**

| Attributes | Deontic | aIm | Conditions | Or else | Type |
|---|---|---|---|---|---|
| **Households** | | Install utilities in higher storeys | If they live in single-family houses | | Shared strategy |
| **Households** | | Implement flood adapted interior fittings | If they live in bungalows and IBA buildings | | Shared strategy |
| **Households** | | Implement flood barriers | If they live in garden houses, apartments and high-rise buildings | | Shared strategy |





| | | | | |
|---|---|---|---|---|
| **Households** | | implement adapted furnishing as a secondary measure | If they have already implemented a measure and if they do not live in bungalows and garden houses | Shared strategy |
| **Authority** | may | Provide subsidies to households to implement measures | e.g., if houses are flooded | Norm |

**Table 2. List of model input factors and their base values.**





| | Model input factors | Symbol | Base Values[a] | Remark |
|---|---|---|---|---|
| **Initial conditions and parameters** | Initial percentage of households with $FE$ | $FEthreshold_{ini}$ | 14% | Based on 2011 census data (age group) and the last major flood in Wilhelmsburg |
| | Initial percentage of households with $CC\ Yes$ | $CCthreshold1_{ini}$ | 44%[b] | Based on NatCen Social Research, 2017 |
| | Initial percentage of households with $CC\ Uncertain$ | $CCthreshold2_{ini}$ | 42%[b] | Based on NatCen Social Research, 2017 |
| | $CC$ update interval (years) | $Y_{CC}$ | 3 | Authors estimation[d] |
| | $SoI$ | $SoI_{ini}$ | 80% | Authors estimation[d] |
| | $SoI$ update interval (years) | $Y_{SoI}$ | 5 | Authors estimation[d] |
| | Initial percentage of $HO\ Own$ | $HO_{ini}$ | 15% | Based on 2011 census data (apartments according to use) |
| | House ownership update | $HO_{update}$ | 1% | Authors estimation[d] |
| | Initial $HI\ Low$ | $HI_{ini}$ | 30% | Authors estimation[d] |
| | Household income update | $HI_{update}$ | 1% | Authors estimation[d] |
| | Abandon frequency threshold | $f_{abandoning}$ | 2 | Authors estimation[d] |
| **Factors for setting up model experiment** | State subsidy | $SS_{lever}$ | 1[c] | Authors estimation[d] |
| | Shared strategy parameter | $SSP$ | 80% | Authors estimation[d] |
| | $SN$ threshold | $SN_{threshold}$ | 30% | Authors estimation[d] |
| | Flood event scenario | $FE_{scenario}$ | Scenario 1 | Authors estimation[d] |
| | Delay parameter (years) | $Y_{delay}$ | 1 | Authors estimation[d] |
| | Adaptation duration (years) | $Y_{adaptation}$ | 7 | Authors estimation[d] |
| | Secondary measure parameter | $SMP$ | 30% | Authors estimation[d] |

[a] The percentage base values are respective to the total number of agents.

[b] The sum of the two $CC$ thresholds should not exceed 100%. If the sum is less than 100%, the remaining is the percentage of agents who do not perceive $CC$ as a source of threat.

[c] $SS_{lever} = 1$ refers to no subsidy.

[d] These estimations are based on authors expertise and knowledge of the study area.





**Table 3. Input factors for model experimentation and their value ranges. Some factors' values are converted from percentages to decimals.**

| Symbol | Range | Step |
|---|---|---|
| $SS_{lever}$ | [1, 3] | 1 |
| $SSP$ | [0.5, 1] | 0.1 |
| $SN_{threshold}$ | [0.2, 0.5] | 0.1 |
| $FE_{scenario}$ | [1, 6] | 1 |
| $Y_{delay}$ | [1, 10] | 2 |
| $Y_{adaptation}$ | [3, 11] | 2 |
| $SMP$ | [0, 0.6] | 0.2 |

**Table C-1. Coefficient of variations ($c_v$) of response factors per iterations. The grey shaded area shows the number of runs in which the $c_v$'s of all the response factors are stable for a difference criterion of 0.001.**

| Response factors | $c_v$ per number of runs | | | | | | | |
|---|---|---|---|---|---|---|---|---|
| | 100 | 500 | 1000 | 1500 | 2000 | 3000 | 4000 | 5000 |
| $Coping_{Yes}$ | 0.015 | 0.034 | 0.029 | 0.027 | 0.027 | 0.027 | 0.027 | 0.027 |
| $Coping_{Yes,SN}$ | 0.024 | 0.052 | 0.045 | 0.041 | 0.041 | 0.043 | 0.042 | 0.043 |
| $PM_{implemented}$ | 0.015 | 0.034 | 0.029 | 0.027 | 0.027 | 0.027 | 0.027 | 0.027 |
| $PM_{abandoned}$ | 0.163 | 0.171 | 0.17 | 0.17 | 0.169 | 0.169 | 0.169 | 0.168 |
| $SM_{implemented}$ | 0.066 | 0.073 | 0.066 | 0.065 | 0.066 | 0.067 | 0.067 | 0.067 |
| $SM_{abandoned}$ | 0.23 | 0.226 | 0.222 | 0.225 | 0.226 | 0.226 | 0.227 | 0.227 |


**Table D-1. Input factors considered in the sensitivity analysis, their distributions and value ranges. Factors specified in percentages are converted to decimals.**

| SA factors | Distribution | Range |
|---|---|---|
| $FEthreshold_{ini}$ | Uniform | [0, 0.3] |
| $CCthreshold_{ini}$ | Discrete | [1, 4][a] |
| $Y_{CC}$ | Discrete | [2, 8] |
| $SoI_{ini}$ | Uniform | [0.5, 1] |
| $Y_{SoI}$ | Discrete | [3, 6] |



| | | |
|---|---|---|
| $HO_{ini}$ | Uniform | [0.1, 0.5] |
| $HO_{update}$ | Uniform | [0, 0.02] |
| $HI_{ini}$ | Uniform | [0.1, 0.5] |
| $HI_{update}$ | Uniform | [0, 0.02] |
| $f_{abandoning}$ | Discrete | [1, 4] |

[a] the $CCthreshold_{ini}$ values for "Yes" and "Uncertain" are 0.35, 0.4, 0.45 and 0.5 for the discrete values of 1, 2, 3 and 4 respectively.
