# Peer review of "The role of household adaptation measures to reduce vulnerability to flooding: a coupled agent-based and flood modelling approach"

_Hydrology and Earth System Sciences, 2020_

## Referee Comment (RC1) · Anonymous Referee #1 · 19 Aug 2020

This manuscript presents an agent-based model (ABM) to examine the role of household adaptation measures in flood risk management (FRM). The research gaps addressed are the inclusion of changes in household behaviour over time and the inclusion economic incentives [line 43]. The stated practical purpose of the model is to inform authorities and communities of the benefits of household adaption measures [l. 440]. The ABM considers threat appraisal (flood experience, perception of climate change) and coping appraisal (e.g. household income, social networks), which determine, through a decision rule table, whether measures will be implemented.

[Figure]

The paper is well written and of interest given the potential of household adaptation measures to help in reducing economic damage under a future potentially more variable climate. The authors should be commended for providing the model, data sets and documentation via github.

A number of sensitivity analyses are performed by modifying given parameters that were initially based on expert opinion [Table 2; Figs 8-12]. The rank of the results, in terms of adaptation uptake and building losses, are broadly as expected given the direction and magnitude of a parameter change. For example, implementation of measures is primarily driven by flood experience, and subsequently delaying implementation makes a community less prepared for the next event [Fig 12]. Considering the results, could the authors comment on the benefits of using the sophisticated ABM approach compared to simpler methods? The challenge is going from sensitivity studies to scenarios. For example, it would be of interest to explore proactive, rather than reactive strategies, examining the role of media on the uptake of measures to inform policy (although this goes beyond the current research).

Minor comments

Figure 7: I can only see 5 lines on this figure (either scenario 1 or 6 is missing or they overlap – it is difficult to differentiate between the colours). Also, it would be expected that there would be a jump in measures implementation following a flood; why is there no jump in year 2 for scenario 2?

Line 331: I do not understand the sentence starting "An important aspect...".

---

## Referee Comment (RC2) · Anonymous Referee #2 · 23 Aug 2020

1. This paper is a follow up to research studies (Abebe et al., 2019a &b) that employ the coupled flood-agent-institution modelling (CLAIM) framework to model the interaction of human with physical flood system in urban environment setting. 2. The novel contribution of this paper is to introduce a new concept of individual behavioral model (Protection Motivation Theory PMT) in exploring the key factors that attributes to household decision making in appraising flood threats and motivations for decision making at the individual level. 3. While the concept itself seems to be innovative and warrant publications, the following reservations/concerns are made:

a. The CLAIM framework as introduced by Abebe et al., (2019a), did not consider the interaction between individual agents and their feedback loop or mechanism. It assumes that agents change interact with the environment and their behavior is greatly influenced by the institutions and their past exposure to flooding. While this could represent the key attributes that impact or influence individual agent's behavior, the role of micro-level agent interactions with each other seems to be ignored. b. The threat appraisal and coping appraisal as presented in the decision trees (Figures 4 & 5), seems to provide level of rationality and control in agent behavior that far from reality and customized around predict and control. The are no feedback loop in the decision trees on figures 4 &5 and the processes are assumed causative and linear. I would argue human behavior is far messier than following coupled of few trajectories in decision making. The role of social network cannot be a consider as external factor as the process of social learning is part of the complex dynamics of interaction between models. c. Having studied the threat appraisal and coping appraisal as shown on Figures 4 and 5, I would clearly be able predict the behavior of the model without a need for a mathematical simulation. This is quite evident from the results as there are few key factors that drive the results. these are: (i) the design of scenarios and the sequencing of storm events; (ii) household past experience to flooding; and (iii) the role of subsidies in the decision making. d. The institutions as defined in Table 1 (shared strategies) seems to be oversimplification of the reality which make it hard to generalize the results and make it more specific to the case under consideration. In the US the role of formal institutions as example Floodplain regulations and penalties associated with nonconformance played significant role in the decision making at a household level. Also Flood Insurance and Flood Rating as part of risk hazard played significant role in how household appraise threats that could be fundamentally different from the threat appraisal action and Coping actions as discussed in this tree. Another assumption that the Source of information as provided by government agencies (levees and flood wall provide protection) is highly subjective and debatable. I would argue being part of the flood managers in one of

the US localities, we are sending different message to our citizens on not relying on structural measures. e. It seems that the PhD thesis (Birkholz, 2014) and structured survey that was undertaken as part of this greatly inform the conceptualization of PMT (Threat and coping appraisals). Hence there should be more of elaboration to link this study with the work of Birkholz. This could be in a form of appendix if the authors believe it would crowd the paper. 4. Having outlined the key concerns , I still believe that the present paper with technical corrections and acknowledgement to the limitations and assumptions discussed above, presented a novel concept and ideas that warrant publication. The design of experiment is adequate, the level of simulations and presentation of results are sufficient and complete. The conclusions researched is substantial and would motivate other research to carry further research. In overall the layout and presentation of the paper is well structured and clear.

Please also note the supplement to this comment:
https://hess.copernicus.org/preprints/hess-2020-272/hess-2020-272-RC2-supplement.pdf

---

## Author Comment (AC1) · 23 Sep 2020

The authors would like to thank Reviewer #1 for taking the time to read the manuscript and provide valuable comments and constructive suggestions. We provide our response to the comments and questions posed by Reviewer #1 in the attached supplement.

---

## Author Comment (AC2) · 24 Sep 2020

**Response to Reviewer #2**

**The authors would like to thank Reviewer #2 for taking the time to read the manuscript and provide valuable comments and constructive suggestions. We address the comments (*our ACs in italics*) follow the comments.**

While the concept itself seems to be innovative and warrant publications, the following reservations/concerns are made:

a. The CLAIM framework as introduced by Abebe et al., (2019a), did not consider the interaction between individual agents and their feedback loop or mechanism. It assumes that agents change interact with the environment and their behavior is greatly influenced by the institutions and their past exposure to flooding. While this could represent the key attributes that impact or influence individual agent's behavior, the role of micro-level agent interactions with each other seems to be ignored.

*AC: Indeed, agents' interaction is a crucial element of the CLAIM framework. In Figure 1 of (Abebe et al., 2019a), the arrow at the top left corner of the **Agents** box, which goes out and enters to the same box, represents that interaction. We acknowledge that there can be numerous actor interactions that influence one another's behaviour although not all interactions are two-ways (with feedbacks). In model conceptualization, the focus is usually on the dominant interactions that affect the behaviour of most agents as it might be neither practical nor relevant to model all the interactions. In the current manuscript, the social network factor (Lines 160-162) is introduced to show household agents' interactions with their neighbours, and how that influence their protection motivation behaviour. In Lines 311-315, we describe that an agent's decision to develop a coping behaviour is influenced by the number of household agents who implemented a measure. This interaction shows an agent's behaviour can be affected by its neighbours, and its behaviour further affects another agent's behaviour. The state subsidy factor also reflects the interaction between the household agents and the authority agent. However, in our conceptualization, that interaction is one-way (from the authority agent to households).*

b. The threat appraisal and coping appraisal as presented in the decision trees (Figures 4 & 5), seems to provide level of rationality and control in agent behavior that far from reality and customized around predict and control. The are no feedback loop in the decision trees on figures 4 &5 and the processes are assumed causative and linear. I would argue human behaviour is far messier than following coupled of few trajectories in decision making. The role of social network cannot be a consider as external factor as the process of social learning is part of the complex dynamics of interaction between models.

*AC: We fully agree with the Reviewer that the rule-based decision trees for agents' threat and coping appraisals are simplified and linear. Human behaviour is complex and challenging to model as many factors play a role for an individual to develop a particular behaviour. Hence, assumptions and abstractions are inevitable in modelling. We used previous empirical studies to define the important factors that affect the threat and coping appraisals of individuals (see Section 3). Some aspects regarding feedback loops can be explained based on the assumptions made. For example, we assumed that (Assumption 13): "If a house has already appraised coping and implemented a measure, they don't appraise coping again, unless they abandon the measure, assuming that they do not implement another primary measure." That means, if an agent already implemented a primary measure, there is no need to go back and update the factors that lead to that decision. Some of these factors (for example, household income and house ownership) are not even affected by the adaptation behaviour of the household.*

*In general, we acknowledge that there are limitations regarding the threat and coping appraisal method used in our modelling. We will address those limitations in the "Discussion and conclusion" section of the revised manuscript.*

c. Having studied the threat appraisal and coping appraisal as shown on Figures 4 and 5, I would clearly be able predict the behaviour of the model without a need for a mathematical simulation. This is quite evident from the results as there are few key factors that drive the results. these are: (i) the design of scenarios and the sequencing of storm events; (ii) household past experience to flooding; and (iii) the role of subsidies in the decision making.

> AC: We applied the protection motivation theory to investigate household-level decision making, but we simplified how we modelled the threat and coping appraisals. The linear and deterministic nature of the decision trees may contribute to the predictability of some of the findings, especially the general trend. However, there are several stochastic elements that could have led to unexpected results. Having said that, we believe it is worth building such models as they could be used to explore other influencing factors such as the role of media in agents protection motivation behaviour (as suggested by Reviewr #1). Future researches may also use intelligent decision-making models such as Bayesian Networks as in (Abdulkareem et al., 2018).
>
> We will address this limitation in the "Discussion and conclusion" section in the revised manuscript.

d. The institutions as defined in Table 1 (shared strategies) seems to be oversimplification of the reality which make it hard to generalize the results and make it more specific to the case under consideration. In the US the role of formal institutions as example Floodplain regulations and penalties associated with nonconformance played significant role in the decision making at a household level. Also Flood Insurance and Flood Rating as part of risk hazard played significant role in how household appraise threats that could be fundamentally different from the threat appraisal action and Coping actions as discussed in this tree. Another assumption that the Source of information as provided by government agencies (levees and flood wall provide protection) is highly subjective and debatable. I would argue being part of the flood managers in one of the US localities, we are sending different message to our citizens on not relying on structural measures.

> AC: The reason we formulated the institutions as shared strategies (not as rules, for example) is to give agents an option whether to develop a protection motivation behaviour or not. Unfortunately, there are no formal institutions in the study area that oblige households to implement any adaptation measure. Hence, we assumed, introducing institutions as shared strategies would be a reasonable starting point for the study area. As described in Lines 222-226, the five institutions we defined are purely hypothetical and are specific to the case as the Reviewer noted. Agent attributes, initial conditions and assumptions defined (conceptualized) in the model are also specific to the case. Hence, the results are not generalizable. However, the modelling approach can be applied to any case to test FRM policy levers and strategies considering heterogeneous individual behaviours (Lines 595-598). Flood insurance is also not available in Wilhelmsburg (Birkholz, 2014, p. 169). That is the reason insurance is not included in the conceptualization. In the "Discussion and conclusion" section of the revised manuscript, we will specifically mention that the institutions are hypothetical. We will also explain why we formulated the institutions as shared strategies.
>
> Regarding the information promoted by the government agencies, unfortunately, they do not advise households to implement individual adaptation measures. Our knowledge of the case study area and informal discussions with authorities are the basis for our assumption. Restemeyer et al. (2015) also discussed the lack of political capital towards realizing individual adaptation measures and considering dikes as "the best way to protect the city and its people" (see p. 54-55).

e. It seems that the PhD thesis (Birkholz, 2014) and structured survey that was undertaken as part of this greatly inform the conceptualization of PMT (Threat and coping appraisals). Hence there should be more of elaboration to link this study with the work of Birkholz. This could be in a form of appendix if the authors believe it would crowd the paper.

*AC: We will add an appendix to elaborate on how (Birkholz, 2014) and the current manuscript are linked.*

*Reference*

*Abdulkareem, S. A., Augustijn, E. W., Mustafa, Y. T. and Filatova, T.: Intelligent judgements over health risks in a spatial agent-based model, International Journal of Health Geographics, 17(1), 8, doi:10.1186/s12942-018-0128-x, 2018.*

---

## Author Comment (AC3) · 24 Sep 2020

**Response to Reviewer #1**

**The authors would like to thank Reviewer #1 for taking the time to read the manuscript and provide valuable comments and constructive suggestions. We address the comments (*our responses in italics*) follow the comments.**

A number of sensitivity analyses are performed by modifying given parameters that were initially based on expert opinion [Table 2; Figs 8-12]. The rank of the results, in terms of adaptation uptake and building losses, are broadly as expected given the direction and magnitude of a parameter change. For example, implementation of measures is primarily driven by flood experience, and subsequently delaying implementation makes a community less prepared for the next event [Fig 12]. Considering the results, could the authors comment on the benefits of using the sophisticated ABM approach compared to simpler methods? The challenge is going from sensitivity studies to scenarios. For example, it would be of interest to explore proactive, rather than reactive strategies, examining the role of media on the uptake of measures to inform policy (although this goes beyond the current research).

*AC: As mentioned in Lines 401-405, we conducted sensitivity analysis only on the initial conditions and parameters as varying these factors is not of interest for the study. See Table 2 for the factors, and Appendix D and Figure D-1 for the SA result. We carried out model experimentation (or scenario analysis) on the second group of factors mentioned in Table 2.*

*Regarding the general benefits of using ABMs compared to simpler methods, we have discussed that in a previous publication (see Abebe et al. 2019a, p. 483-485). In the current study, we applied the protection motivation theory to investigate household-level decision making, but we simplified how we modelled the threat and coping appraisals. We used decision trees that are somehow deterministic and linear, which may have contributed to the predictability of some model results. The social network is also modelled in a simplified manner, which could be improved with more data. However, there are several stochastic elements that could have led to unexpected results. Future researches may use intelligent decision-making models such as Bayesian Networks as in (Abdulkareem et al., 2018). In addition, as the Reviewer commented, model conceptualization could include institutions that govern proactive strategies and the role of media in agents' protection motivation behaviour if that is of particular interest. The current manuscript aims to provoke communities and decision-makers in the study area to investigate further the role of household adaptation measures in mitigating potential flood damages.*

*We will address this limitation in the "Discussion and conclusion" section in the revised manuscript.*

Figure 7: I can only see 5 lines on this figure (either scenario 1 or 6 is missing or they overlap – it is difficult to differentiate between the colours). Also, it would be expected that there would be a jump in measures implementation following a flood; why is there no jump in year 2 for scenario 2?

*AC: We assume that the Reviewer meant to comment on Figure 9. As we mentioned in Line 461, the curves appear to overlap are that of Scenario 1 and Scenario 4. The reason there is no big jump in the number of houses that implemented primary measures in year 2 for Scenario 2 is that flood event B is a small event, and it only affects a few houses. Hence, its effect on the number of primary measures is minimal (but not zero). The line appears flat but zooming in at year 2 shows there is a minor change in the slope of the curve.*

*We will add explanations concerning the overlapping curves and why there is no big jump in year 2 for Scenario 2 in the revised manuscript.*

Line 331: I do not understand the sentence starting "An important aspect...".

*AC: As mentioned in Lines 235-237, when a shared strategy drives a system, agents do what the majority in that system does. However, the household also has the option not to implement the measure without incurring any punishment. In our conceptualization, the SN factor is the same for all households who live in a similar house category. That means, if the value of the SN factor is "High", all households who live in that house category will follow the same behaviour. But, as discussed above, households have the option not to develop that behaviour though most follow the crowd. To reflect this property of shared strategies, we introduced another factor, the shared strategy parameter (SSP). The SSP is a kind of threshold that defines the percentage of household agents that follow the shared strategy. For every agent, if the SN is High, and a randomly drawn number (from a uniform distribution) is greater than the SSP value, the agent develops the intended behaviour (i.e., the coping behaviour).*

*We will improve the referred sentences according to the above explanation in the revised manuscript.*

*Reference*

*Abdulkareem, S. A., Augustijn, E. W., Mustafa, Y. T. and Filatova, T.: Intelligent judgements over health risks in a spatial agent-based model, International Journal of Health Geographics, 17(1), 8, doi:10.1186/s12942-018-0128-x, 2018.*